# Structured Diffusion Models with Mixture of Gaussians as Prior Distribution

## Abstract

We propose a class of structured diffusion models, in which the prior distribution is chosen as a mixture of Gaussians, rather than a standard Gaussian distribution. The specific mixed Gaussian distribution, as prior, can be chosen to incorporate certain structured information of the data. We develop a simple-to-implement training procedure that smoothly accommodates the use of mixed Gaussian as prior. Theory is provided to quantify the benefits of our proposed models, compared to the classical diffusion models. Numerical experiments with synthetic, image and operational data are conducted to show comparative advantages of our model. Our method is shown to be robust to mis-specifications and in particular suits situations where training resources are limited or faster training in real time is desired.

## 1 Introduction

Diffusion models, since Ho et al. (2020); Song et al. (2020b), have soon emerged as a powerful class of generative models to handle the training and generation for various forms of contents, such as image and audio. On top of the success, like many other models, training a diffusion model can require significant computational resources. Compared to more classical generative models such as generative adversarial networks (GAN), the inherent structure of diffusion models requires multiple steps to gradually corrupt structured data into noise and then reverse this process. This necessitates a large number of training steps to successfully denoise, further adding to the computational cost associated with the network and data size.

That said, not all scenarios where diffusion models are used enjoy access to extensive training resources. For example, small-sized or non-profit enterprises with limited budget of compute may desire to train diffusion models with their private data. In those cases with limited resources, the training of standard diffusion models may encounter budget challenges and cannot afford the training of adequate number of steps. In addition, there are scenarios one desires to train in real time with streaming data and aims at achieving certain training performance level as fast as possible given a fixed amount of resources. In such cases, it is preferable to further improve classical diffusion models to achieve faster training.

If training resources are limited, insufficient training can hinder the performance of the diffusion models and result in poorly generated samples. Below in Figure 1 is an illustrative example on grayscale digits images, showing the performance of denoising diffusion probabilistic models given different training steps. When the model is trained for only 800 steps, it has limited exposure to the data, and as a result, the generated samples are likely to be blurry, incomplete, or show a lack of consistency in terms of digit shapes and structures. The model at this stage has not yet learned to fully reverse the noise process effectively. Our work was motivated by the considerations to improve training efficiency so that, if resources are limited, even with fewer training steps one can achieve certain satisfying level of performance.

In this work, we aim to provide one approach based on adjusting the prior distribution, to improve the performances of classical diffusion models when training resources are limited. Classical diffusion models use Gaussian distribution as the prior distribution, which was designed due to the manifold hypothesis and sampling in low-density areas Song & Ermon (2019). However, this approach does not use the potential structured information within the data and considerably adds to the training complexity. Admittedly, when training resources are not a constraint, or when the data structure is

| (a) 0.8k steps | (b) 1.6k steps | (c) 2.4k steps | (d) 3.2k steps |

Figure 1: DDPM with varying training steps

difficult to interpret, the use of Gaussian distribution as a prior can be a safe and decent choice. That said, when users have certain structured domain knowledge about the data, say, there might be some clustered groups of data on some dimensions, it can be useful to integrate such information into the training of diffusion models. To increase the ability to incorporate such information, we propose the use of a Gaussian Mixture Model (GMM) as the prior distribution. We develop the associated training process, and examine the comparative performances. The main results of our work are summarized as follows.

1) We propose a class of mixed diffusion models, whose prior distribution is Gaussian mixture model instead of Gaussian noise in the previous diffusion literature. We detail the forward process and the reverse process of the mixed diffusion models, including both the Mixed Denoising Diffusion Probabilistic Models (mixDDPM) and the Mixed Score-based Generative Models (mixSGM) with an auxiliary dispatcher that assigns each data to their corresponding center.

2) We introduce a quantative metric "Reverse Effort" in the reverse process, which measures the distance between the prior distribution and the finite-sample data distribution under appropriate coupling. With the 'Reverse Effort', we explain the benefits of the mixed diffusion models by quantifying the effort-reduction effect, which further substantiates the efficiency of this mixed model.

3) We conduct various numerical experiments among synthesized datasets, operational datasets and image datasets. All numerical results have advocated the efficiency of the mixed diffusion models, especially in the case when training resources are limited.

## 1.1 RELATED LITERATURE

**Diffusion models and analysis.** Diffusion models work by modeling the process of data degradation and subsequently reversing this process to generate new samples from noise. The success of diffusion models lies in their ability to generate high-quality, diverse outputs. Their application has expanded across fields such as image and audio synthesis tasks Kong et al. (2020); Dhariwal & Nichol (2021); Leng et al. (2022); Rombach et al. (2022); Yu et al. (2024); Skorokhodov et al. (2024), image editing Meng et al. (2021); Avrahami et al. (2022); Kawar et al. (2023); Mokady et al. (2023), text-to-image generation Saharia et al. (2022); Zhang et al. (2023); Kawar et al. (2023), and other downstream tasks including in medical image generation Khader et al. (2023); Kazerouni et al. (2023) and modeling molecular dynamics Wu & Li (2023); Arts et al. (2023), making them a pivotal innovation in the landscape of generative AI. Tang & Zhao (2024) provide further understanding of score-based diffusion models via stochastic differential equations.

**Other methods for efficiency improvement.** Various literature have contributed to improve the performance of the diffusion models by proposing more efficient noise schedules Kingma et al. (2021); Karras et al. (2022); Hang & Gu (2024), introducing latent structures Rombach et al. (2022); Kim et al. (2023); Podell et al. (2024); Pernias et al. (2024), improving training efficiency Wang et al. (2023); Haxholli & Lorenzi (2023) and applying faster samplers Song et al. (2020a); Lu et al. (2022a;b); Watson et al. (2022); Zhang & Chen (2023); Zheng et al. (2023); Pandey et al. (2024b); Xue et al. (2024); Zhao et al. (2024); Guo et al. (2024). In addition, Yang et al. (2024) employs a spectrum of neural networks whose sizes are adapted according to the importance of each generative step.

**Use of non-Gaussian prior distribution.** There exists a series of related but different work on using non-Gaussian noise distributions, to enhance the performance and efficiency of the diffusion models; see Nachmani et al. (2021); Zach et al. (2023); Yen et al. (2023); Bansal et al. (2024); Pandey et al. (2024a), among others. Our work instead emphasizes on the use of structured prior distribution (instead of noise distribution), with the purpose to focus on incorporating data information into the model.

The following of this paper are organized as follows. Section 2 reviews the background of diffusion models, including both Denoising Diffusion Probabilistic Model Ho et al. (2020) and Score-based Generative Models Song et al. (2020b). Section 3 starts from numerical experiments on 1D syntatic datasets to illustrate the motivation of this work. Section 4 details our new models and provide theoretical analysis. Section 5 includes numerical experiments and Section 6 concludes the paper with future directions.

## 2    BRIEF REVIEW ON DIFFUSION MODELS AND NOTATION

In this section, we briefly review the two classical class of diffusion models: Denoising Diffusion Probabilistic Model (DDPM) Ho et al. (2020) in Section 2.1, and Score-based Generative Models (SGM) Song et al. (2020b) in Section 2.2. Meanwhile, we specify the notation related to the diffusion models and prepare for the description of our proposed methods later in Section 4.

### 2.1    DENOISING DIFFUSION PROBABILISTIC MODEL

In DDPM, the forward process is modeled as a discrete-time Markov chain with Gaussian transition kernels. This Markov chain starts with the observed data $\mathbf{x}_0$, which follows the data distribution $p_{\text{data}}$. The forward process gradually adds noises to $\mathbf{x}_0$ and forms a finite-time Markov process $\{\mathbf{x}_0, \mathbf{x}_1, \cdots, \mathbf{x}_T\}$. The transition density of this Markov chain can be written as

$$\mathbf{x}_t | \mathbf{x}_{t-1} \sim \mathcal{N} \left( \sqrt{1 - \beta_t} \mathbf{x}_{t-1}, \beta_t \boldsymbol{I} \right), \quad t = 1, 2, \cdots, T, \tag{1}$$

where $\beta_1, \cdots, \beta_T$ is called the noise schedule. Then, the marginal density of $\mathbf{x}_t$ conditional on $x_0$ can be written in closed-form: $\mathbf{x}_t | \mathbf{x}_0 \sim \mathcal{N} \left( \sqrt{\overline{\alpha}_t} \mathbf{x}_0, (1 - \overline{\alpha}_t) \boldsymbol{I} \right)$, where $\overline{\alpha}_t = \prod_{s=1}^{t} (1 - \beta_s)$ for $t = 1, 2, \cdots, T$. The noise schedule is chosen so that $\alpha_T$ is closet to 0.

During the training process, a neural network $\epsilon_\theta : \mathbb{R}^d \times \{1, 2, \cdots, T\} \to \mathbb{R}^d$ parameterized by $\theta$ is trained to predict the random Gaussian noise $\epsilon$ given the time $t$ and the value of the forward process $\mathbf{x}_t$. The DDPM training objective is proposed as

$$\mathcal{L}^{\text{DDPM}} := \sum_{t=1}^{T} \mathbb{E}_{\mathbf{x}_0, \epsilon} \left[ \omega_t \cdot \| \epsilon - \epsilon_\theta \left( \mathbf{x}_t, t \right) \|_2^2 \right] \text{ with } \mathbf{x}_t = \sqrt{\overline{\alpha}_t} \mathbf{x}_0 + (1 - \overline{\alpha}_t) \epsilon \tag{2}$$

where $\omega_1, \omega_2, \cdots, \omega_T$ is a sequence of weights and $\| \cdot \|_2$ is the $l_2$ metric.

Based on the trained neural network, the reverse sampling process is also modeled as a discrete-time process with Gaussian transition kernels. Here and throughout what follows, we use $\tilde{\mathbf{x}}$ to denote the reverse process. The reverse process starts from the prior distribution $\tilde{\mathbf{x}}_T \sim \mathcal{N}(\mathbf{0}, \boldsymbol{I})$ and the transition density is given by

$$\tilde{\mathbf{x}}_{t-1} | \tilde{\mathbf{x}}_t \sim \mathcal{N}(\boldsymbol{\mu}_\theta(\tilde{\mathbf{x}}_t, t), \beta_t \boldsymbol{I}) \text{ with } \boldsymbol{\mu}_\theta(\mathbf{x}, t) = \frac{1}{\sqrt{1 - \beta_t}} \left( \mathbf{x} - \frac{\beta_t}{\sqrt{1 - \overline{\alpha}_t}} \epsilon_\theta(\mathbf{x}, t) \right) \tag{3}$$

for $t = 1, 2, \cdots, T$. The final result $\tilde{\mathbf{x}}_0$ is considered to be the output of the DDPM and its distribution is used to approximate the data distribution $p_{\text{data}}$.

### 2.2    SCORE-BASED GENERATIVE MODELS

Both of the forward process and the reverse process of the SGM are modeled by Stochastic Differential Equations (SDE). The forward SDE starts from $\mathbf{x}_0 \sim p_{\text{data}}$ and evolves according to

$$d\mathbf{x}_t = f_t \mathbf{x}_t dt + g_t d\mathbf{w}_t, \tag{4}$$

where $f_t$ is the drift scalar function, $g_t$ is the diffusion scalar function and $\mathbf{w}_t$ is a d-dimensional standard Brownian motion. In particular, the forward SDE is an Ornstein–Uhlenbeck (OU) process and the marginal distribution has a closed-form Gaussian representation. Without loss of generality, we suppose $\mathbf{x}_t \sim \mathcal{N}(\alpha_t \mathbf{x}_0, \sigma_t^2 \boldsymbol{I})$, where $\alpha_t$ and $\sigma_t$ are solely determined by the scalar functions $f_t, g_t$.

According to Anderson (1982), the reverse of diffusion process (4) is also a diffusion process and can be represented as

$$d\tilde{\mathbf{x}}_t = \left( f_t \tilde{\mathbf{x}}_t - g_t^2 \nabla_{\mathbf{x}} \log p_t(\tilde{\mathbf{x}}_t) \right) dt + g_t d\tilde{\mathbf{w}}_t, \quad \tilde{\mathbf{x}}_T \sim \mathcal{N}(\mathbf{0}, \sigma_T^2 \boldsymbol{I}), \tag{5}$$

where $dt$ is an infinitesimal negative time step and $\tilde{dw}$ is the standard Brownian motion when time flows back from $T$ to $0$. Besides, $\nabla_{\mathbf{x}} \log p_t(\mathbf{x})$ is called the score function and is approximated by a trained neural network $\boldsymbol{s}_\theta$. However, later researchers have substituted the score function by the noise model Lu et al. (2022a) and the prediction model Lu et al. (2022b) to improve the overall efficiency of the SGM. To keep align with the DDPM, we only introduce the noise model here.

Instead of learning the score function directly, the noise model utilizes a neural network $\epsilon_\theta(\mathbf{x}, t) : \mathbb{R}^d \times (0, T] \to \mathbb{R}^d$ to learn the scaled score function $-\sigma_t \nabla_{\mathbf{x}} \log p_t(\mathbf{x})$. According to Lu et al. (2022a), the training objective is elected to be

$$\mathcal{L}^{\text{SGM}} := \int_0^T \omega_t \cdot \mathbb{E}_{\mathbf{x}_0, \epsilon} \left[ \| \epsilon_\theta(\mathbf{x}_t, t) - \epsilon \|_2^2 \right] dt \quad \text{with} \quad \mathbf{x}_t = \alpha_t \mathbf{x}_0 + \sigma_t \epsilon, \tag{6}$$

where $\omega_t$ is a weighting function. Having the trained noise model $\epsilon_\theta$, the previous reverse SDE (5) can be re-formalized as

$$d\tilde{\mathbf{x}}_t = \left( f_t \tilde{\mathbf{x}}_t + \frac{g_t^2}{\sigma_t} \epsilon_\theta(\tilde{\mathbf{x}}_t, t) \right) dt + g_t d\tilde{\mathbf{w}}_t, \quad \tilde{\mathbf{x}}_T \sim \mathcal{N}(\mathbf{0}, \sigma_T^2 \boldsymbol{I}). \tag{7}$$

Various numerical SDE solvers can be applied on (7) to obtain the final output $\tilde{\mathbf{x}}_0$.

# 3 ILLUSTRATION WITH ONE-DIMENSIONAL EXAMPLES

In this section, we provide a brief numerical illustration to show the performance comparison between DDPM and mixDDPM (the method that will be formally introduced in the next section). We illustrate through two 1-dimensional experiments. The true data distribution for the first experiment is a standardized Gaussian mixture distribution with symmetric clusters $p_{\text{data}} = \frac{1}{2}(\mathcal{N}(-0.9, 0.19) + \mathcal{N}(0.9, 0.19))$. The second experiment chooses a Gamma mixture distribution as the data distribution that shares the same cluster mean and variance with the above-mentioned Gaussian mixture distribution.

Table 1: DDPM v.s. mixDDPM on 1D Gaussian Mixture Model

|  | DDPM | mixDDPM |
|---|---|---|
| $\mathcal{W}_1$ distance | 0.222 | 0.113 |
| K-S statistics | 0.213 | 0.073 |

Table 2: DDPM v.s. mixDDPM on 1D Gamma Mixture Model

|  | DDPM | mixDDPM |
|---|---|---|
| $\mathcal{W}_1$ distance | 0.206 | 0.136 |
| K-S statistics | 0.232 | 0.103 |

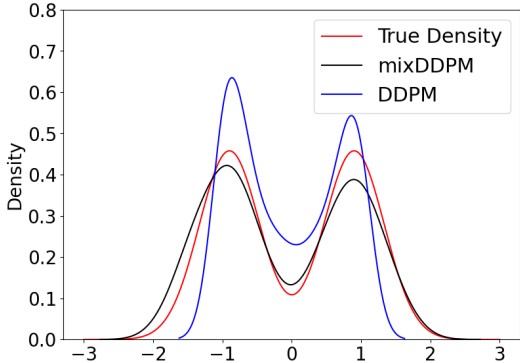

Figure 2: DDPM and mixDDPM on 1D Bimodal Gaussian Mixture Model

We present the results in Table 1 and Table 2. The experiment setting is as follows. The training dataset includes 256 samples from the data distribution. The classical DDPM involves 16k steps.

We also implement the mixDDPM, which is our new model and will be introduced in Section 4, on the training dataset with the same neural network architecture, the same size of network parameters, the same training steps and the same random seeds. We calculate the $\mathcal{W}_1$ distance and the Kolmogorov–Smirnov (K-S) statistics Massey Jr (1951); Fasano & Franceschini (1987); Berger & Zhou (2014) between the finite-sample distributions of the generated samples and the true data distribution. In addition, we draw the density of both the finite-sample distributions and the data distribution for the Gaussian mixture experiment in Figure 2.

This illustration of two 1-dimensional examples shows that when training steps are not adequate, the mixDDPM model with a non-Gaussian prior has the potential to achieve significant better performance than the classical DDPM model, by making sure all else is equal.

## 4 MIXED DIFFUSION MODELS

We propose in this section the mixed diffusion models. Instead of maintaining Gaussian prior distribution, our model chooses the Gaussian mixture model. While still keeping the benefits of Gaussian prior distributions, i.e., the manifold hypothesis and sampling in low density area, the additional parameters enable the model to incorporate more information about the data distribution and further reduce the overall loads of the reverse process. In what follows, we first introduce how to incorporate data information into the prior distribution and an additional dispatcher. We leave more details of the mixDDPM and the mixSGM in Subsection 4.1 and Subsection 4.2, respectively.

In general, the prior distributions in the mixed diffusion models belong to a class of Gaussian mixture distributions with centers $\mathbf{c}_1, \cdots, \mathbf{c}_K$. The number of centers $K$, as well as the specific values of the centers $\mathbf{c}_1, \cdots, \mathbf{c}_K$, are predetermined before training and sampling in our model. These parameters can be flexibly chosen by users of the model, either through domain knowledge, or through various analysis methods. There is no need to concern whether the choices are optimal or not. Instead, such choices only need to contain some partial structure information of the data known by users. For instance, users may employ clustering techniques in some particular low-dimensional spaces of the data or use labels on some dimensions of the data. Additional discussions on possible center-selection methods are provided in the Appendix.

Now for each data sample $\mathbf{x}_0 \sim p_{\text{data}}$, a dispatcher $D : \mathbb{R}^d \to \{1, 2, \cdots, K\}$ assigns $\mathbf{x}_0$ to one of the center. In the context of this work, the dispatcher is defined in the following way:

$$D(\mathbf{x}) = \arg\min_i \{d(\mathbf{x}, \mathbf{c}_i)\}, \tag{8}$$

where $d(\cdot, \cdot)$ is a distance metric. For example, we can set it to be $l_2$ distance. In other words, the dispatcher assigns the data $\mathbf{x}_0$ to the nearest center.

### 4.1 THE MIXDDPM

We first introduce the forward process. Given data sample $\mathbf{x}_0$, we suppose the dispatcher assigns it to the $j$-th center, i.e, $D(\mathbf{x}_0) = j$. Similar to DDPM, the forward process is a discrete-time Markov chain. Conditional on $\mathbf{x}_{t-1}$ and $D(\mathbf{x}_0) = j$, the distribution of $\mathbf{x}_t$ is given by

$$\mathbf{x}_t | \mathbf{x}_{t-1} \sim \mathcal{N}\left(\sqrt{1 - \beta_t}(\mathbf{x}_{t-1} - \mathbf{c}_j) + \mathbf{c}_j, \beta_t \boldsymbol{I}\right), \quad t = 1, 2, \cdots, T. \tag{9}$$

In other words, the process $\mathbf{x}_t - \mathbf{c}_j$ follows the same transition density as (1). As a result, the marginal distribution of $\mathbf{x}_t$, conditional on $\mathbf{x}_0$ assigned to the j-th center, is

$$\mathbf{x}_t | \mathbf{x}_0 \sim \mathcal{N}\left(\sqrt{\overline{\alpha}_t}(\mathbf{x}_0 - \mathbf{c}_j) + \mathbf{c}_j, (1 - \overline{\alpha}_t)\boldsymbol{I}\right), \quad t = 1, 2, \cdots, T. \tag{10}$$

As $t$ increases, the distribution of $\mathbf{x}_t$ gradually converges to $\mathcal{N}(\mathbf{c}_j, \boldsymbol{I})$. That said, the prior distribution conditional on $D(\mathbf{x}_0) = j$ is a Gaussian distribution centered at $\mathbf{c}_j$ with unit variance. Hence, the prior distribution is a Gaussian mixture model that can be represented as $\sum_{i=1}^K p_i \mathcal{N}(\mathbf{c}_i, \boldsymbol{I})$, where $p_i$ is the proportion of data that are assigned to the i-th center.

To learn the noise given the forward process, the mixed DDPM utilizes a neural network $\epsilon_\theta : \mathbb{R}^d \times \{1, 2, \cdots, T\} \times \{1, 2, \cdots, K\} \to \mathbb{R}^d$. The neural network takes three inputs: the state of the forward process $\mathbf{x}_t \in \mathbb{R}^d$, the time $t \in \{1, 2, \cdots, T\}$ and the center number $D \in \{1, 2, \cdots, K\}$.

Our method adopts the U-Net architecture, as suggested by Ho et al. (2020); Rombach et al. (2022); Ramesh et al. (2022). Similar to (2), the mixed DDPM adopts the following training objective:

$$\mathcal{L}_{\mathrm{mix}}^{\mathrm{DDPM}} := \sum_{t=1}^{T} \mathbb{E}_{\mathbf{x}_0,\epsilon} \left[ \omega_t \cdot \|\epsilon - \epsilon_\theta(\mathbf{x}_t, t, j)\|_2^2 \right], \quad \text{with} \quad \mathbf{x}_t = \sqrt{\bar{\alpha}_t}(\mathbf{x}_0 - \mathbf{c}_j) + \mathbf{c}_j + \sqrt{1 - \bar{\alpha}_t}\epsilon.$$

(11)

The training process can be viewed as solving the optimization problem $\min_\theta \mathcal{L}_{\mathrm{mix}}^{\mathrm{DDPM}}$ by the stochastic gradient descent method.

During the reverse sampling process, the mixed DDPM first samples $\tilde{\mathbf{x}}_T \sim \sum_{i=1}^{K} p_i \mathcal{N}(\mathbf{c}_i, \mathbf{I})$. To do so, it first samples $j$ from $\{1, 2, \cdots, K\}$ such that $\mathbb{P}(j = i) = p_i$. Then, it proceeds to sample $\tilde{\mathbf{x}}_T \sim \mathcal{N}(\mathbf{c}_j, \mathbf{I})$. The transition density for $\tilde{\mathbf{x}}_{t-1}$, conditional on $\tilde{\mathbf{x}}_t$, is given by

$$\tilde{\mathbf{x}}_{t-1} | \tilde{\mathbf{x}}_t \sim \mathcal{N}(\boldsymbol{\mu}_\theta(\tilde{\mathbf{x}}_t, t), \beta_t \mathbf{I}), \quad t = 1, 2, \cdots, T,$$

(12)

where

$$\boldsymbol{\mu}_\theta(\mathbf{x}, t) = \frac{1}{\sqrt{1 - \beta_t}} \left( \mathbf{x} - \mathbf{c}_j - \frac{\beta_t}{\sqrt{1 - \bar{\alpha}_t}} \epsilon_\theta(\mathbf{x}, t, j) \right) + \mathbf{c}_j.$$

(13)

We summarize the training process and the sampling process for the mixed DDPM in Algorithm 1 and Algorithm 2 below.

---

**Algorithm 1** Training Process for the mixDDPM

---

**Input:** samples $\mathbf{x}_0$ from the data distribution, un-trained neural network $\epsilon_\theta$, time horizon $T$, noise schedule $\beta_1, \cdots, \beta_T$, number of centers $K$ and the centers $\mathbf{c}_1, \cdots, \mathbf{c}_K$
**Output:** Trained neural network $\epsilon_\theta$
**repeat**
    Get data $\mathbf{x}_0$
    Find center $j = D(\mathbf{x}_0)$
    Sample $t \sim U\{1, 2, \cdots, T\}$ and $\epsilon \sim \mathcal{N}(\mathbf{0}, \mathbf{I})$
    $\mathbf{x}_t \leftarrow \sqrt{\bar{\alpha}_t}(\mathbf{x}_0 - \mathbf{c}_j) + \mathbf{c}_j + \sqrt{1 - \bar{\alpha}_t}\epsilon$
    $\mathcal{L} \leftarrow \omega_t \|\epsilon - \epsilon_\theta(\mathbf{x}_t, t, j)\|_2^2$
    Take a gradient descent step on $\nabla_\theta \mathcal{L}$
**until** Converged or training resource/time limit is hit

---

**Algorithm 2** Reverse Process for the mixDDPM

---

**Input:** Trained neural network $\epsilon_\theta$, center weights $p_1, \cdots, p_K$, centers $\mathbf{c}_1, \cdots, \mathbf{c}_K$
Sample $j \in \{1, \cdots, K\}$ with $\mathbb{P}(j = i) = p_i$ for $i = 1, \cdots, K$
Sample $\tilde{\mathbf{x}}_T \sim \mathcal{N}(\mathbf{c}_j, \mathbf{I})$
**for** $t = T$ to 1 **do**
    Calculate $\boldsymbol{\mu}_\theta(\tilde{\mathbf{x}}_t, t) = \frac{1}{\sqrt{1 - \beta_t}} \left( \tilde{\mathbf{x}}_t - \mathbf{c}_j - \frac{\beta_t}{\sqrt{1 - \bar{\alpha}_t}} \epsilon_\theta(\tilde{\mathbf{x}}_t, t, j) \right) + \mathbf{c}_j$
    Sample $\tilde{\mathbf{x}}_{t-1} \sim \mathcal{N}(\boldsymbol{\mu}_\theta(\tilde{\mathbf{x}}_t, t), \beta_t \mathbf{I})$
**end for**
**Return** $\tilde{\mathbf{x}}_0$

---

Before ending this section, we illustrate why the mixDDPM improves the overall efficiency of the DDPM. We first define the reverse effort for the DDPM and the mixDDPM by

$$\mathrm{ReEff}^{\mathrm{DDPM}} := \mathbb{E}_{\tilde{\mathbf{x}}_T \sim \mathcal{N}(\mathbf{0}, \mathbf{I}),\ \mathbf{x}_0 \sim \bar{p}_{\mathrm{data}}} \left[ \|\mathbf{x}_0 - \tilde{\mathbf{x}}_T\|^2 \right],$$

(14)

$$\mathrm{ReEff}_{\mathrm{mix}}^{\mathrm{DDPM}} := \mathbb{E}_{\tilde{\mathbf{x}}_T \sim \mathcal{N}(\mathbf{c}_{D(\mathbf{x}_0)}, \mathbf{I})} \mathbb{E}_{\mathbf{x}_0 \sim \bar{p}_{\mathrm{data}}} \left[ \|\mathbf{x}_0 - \tilde{\mathbf{x}}_T\|^2 \right],$$

(15)

where $\bar{p}_{\mathrm{data}}$ is the empirical distribution over the given data. We now explain the definition of the reverse effort. The forward process gradually adds noise to the initial data $\mathbf{x}_0$, until it converges to the prior distribution. On the contrary, the reverse process aims to recover $\mathbf{x}_0$ given $\tilde{\mathbf{x}}_T$ as input. Hence, we evaluate the distance between $\mathbf{x}_0$ and $\tilde{\mathbf{x}}_T$ and define its expectation as the reverse effort. One noteworthy fact of the reverse effort for the mixDDPM is that $\mathbf{x}_0$ and $\tilde{\mathbf{x}}_T$ are not independent. This can be attributed to the dispatcher, which assigns $\mathbf{x}_0$ to the $D(\mathbf{x}_0)$-th center. We present the relationship between the two reverse efforts defined by (14) and (15) in Proposition 1.

**Proposition 1.** *Given the cluster number $K$ and the cluster centers $\mathbf{c}_1, \cdots, \mathbf{c}_K$, we define $X_i = \{\mathbf{x} : D(\mathbf{x}) = i\}$ and $p_i = \frac{|X_i|}{\sum_{j=1}^{K}|X_j|}$ for $i = 1, 2, \cdots, K$. Under the assumption that $\mathbf{c}_i$ is the arithmetic mean of $X_i$, we have*

$$\text{ReEff}_{\text{mix}}^{\text{DDPM}} = \text{ReEff}^{\text{DDPM}} - \sum_{i=1}^{K} p_i \|\mathbf{c}_i\|^2. \tag{16}$$

Proposition 1 shows that the mixDDPM requires less reverse effort compared to the classical DDPM. In addition, this reduction can be quantified as a weighted average of the $l_2$-norm of the centers. This reduction can be understood in the following way. We have discussed in Section 3 that the prior distribution of the DDPM contains no information about the data distribution. In contrast, the prior distribution of the mixDDPM retains some data information through the choice of the centers $\mathbf{c}_1, \cdots, \mathbf{c}_K$. This retained data information, together with the dispatcher, helps reduce the reverse effort by providing guidance on where to initiate the reverse process. Although this reduction may not significantly affect sampling quality when the neural network is well-trained, it can lead to potential improvements when training is insufficient.

## 4.2 THE MIXSGM

Suppose a given data $\mathbf{x}_0$ is assigned to the $j$-th center by the dispatcher, i.e, $D(\mathbf{x}_0) = j$. The mixed SGM modifies the forward SDE from (4) to

$$d\mathbf{x}_t = f_t(\mathbf{x}_t - \mathbf{c}_j)dt + g_t d\mathbf{w}_t, \tag{17}$$

Equivalently, $\mathbf{x}_t - \mathbf{c}_j$ is the OU-process that follows (4). Then, the marginal distribution of $\mathbf{x}_t$, conditional on $\mathbf{x}_0$ and $D(\mathbf{x}_0) = j$, can be calculated as $\mathcal{N}(\alpha_t\mathbf{x}_0 + \mathbf{c}_j, \sigma_t^2\mathbf{I})$. As the time horizon $T$ increases, the prior distribution, conditional on $D(\mathbf{x}_0) = j$, is $\mathcal{N}(\alpha_T\mathbf{x}_0 + \mathbf{c}_j, \sigma_T^2\mathbf{I})$, which can be approximated by $\mathcal{N}(\mathbf{c}_j, \sigma_T^2\mathbf{I})$ if $\alpha_T$ is small enough. Hence, the unconditional prior distribution for the mixed SGM is chosen to be $\sum_{i=1}^{K} p_i\mathcal{N}(\mathbf{c}_i, \sigma_T^2\mathbf{I})$, where $p_i$ is the proportion of data that are assigned to the $i$-th center. Below we draw a table to summarize and compare the prior distributions of both classical and mixed diffusion models.

Table 3: Prior distributions

| Prior distribution | DDPM | SGM |
|---|---|---|
| Classical | $\mathcal{N}(\mathbf{0}, \mathbf{I})$ | $\mathcal{N}(\mathbf{0}, \sigma_T^2\mathbf{I})$ |
| Mixed (our model) | $\sum_{i=1}^{K} p_i\mathcal{N}(\mathbf{c}_i, \mathbf{I})$ | $\sum_{i=1}^{K} p_i\mathcal{N}(\mathbf{c}_i, \sigma_T^2\mathbf{I})$ |

Again, we adopt the U-Net architecture to define the noise model $\epsilon_\theta : \mathbb{R}^d \times (0, T) \times \{1, 2, \cdots, K\}$. Following (6), the training process can be modeled as solving the following optimization problem by stochastic gradient descent: $\min_\theta \mathcal{L}_{\text{mix}}^{\text{SGM}}$, where $\mathcal{L}_{\text{mix}}^{\text{SGM}} = \int_0^T \omega_t \cdot \mathbb{E}_{\mathbf{x}_0, \epsilon}\left[\|\epsilon_\theta(\mathbf{x}_t, t, j) - \epsilon\|_2^2\right] dt$ with $\mathbf{x}_t = \alpha_t\mathbf{x}_0 + \mathbf{c}_j + \sigma_t\epsilon$ and $j = D(\mathbf{x}_0)$.

Finally, the reverse sampling process can be modeled as both reverse SDE and probability ODE. Similar to what the mixed DDPM has done in Section 4.1, the mixed SGM first samples $j$ from $\{1, 2, \cdots, K\}$ according to the weights $\mathbb{P}(j = i) = p_i$ and then samples $\tilde{\mathbf{x}}_T \sim \mathcal{N}(\mathbf{c}_j, \mathbf{I})$. The corresponding reverse SDE is given by

$$d\tilde{\mathbf{x}}_t = \left(f_t(\tilde{\mathbf{x}}_t - \mathbf{c}_j) + \frac{g_t^2}{\sigma_t}\epsilon_\theta(\tilde{\mathbf{x}}_t, t, j)\right) dt + g_t d\tilde{\mathbf{w}}_t, \tag{18}$$

For ease of exposition, we present the training process and the sampling process for the mixSGM in Appendix C.2. We also present the following Proposition to illustrate the effort-reduction effect of the mixSGM.

**Proposition 2.** *Define*

$$\text{ReEff}^{\text{SGM}} := \mathbb{E}_{\tilde{\mathbf{x}}_T \sim \mathcal{N}(\mathbf{0}, \sigma_T^2\mathbf{I}),\, \mathbf{x}_0 \sim \bar{p}_{\text{data}}}\left[\|\mathbf{x}_0 - \tilde{\mathbf{x}}_T\|^2\right], \tag{19}$$

$$\text{ReEff}_{\text{mix}}^{\text{SGM}} := \mathbb{E}_{\tilde{\mathbf{x}}_T \sim \mathcal{N}(\mathbf{c}_{D(\mathbf{x}_0)}, \sigma_T^2\mathbf{I})}\mathbb{E}_{\mathbf{x}_0 \sim \bar{p}_{\text{data}}}\left[\|\mathbf{x}_0 - \tilde{\mathbf{x}}_T\|^2\right], \tag{20}$$

where $\bar{p}_{\mathrm{data}}$ is the empirical distribution over the given data Given the cluster number $K$ and the cluster centers $\mathbf{c}_1, \cdots, \mathbf{c}_K$, we define $X_i = \{\mathbf{x} : D(\mathbf{x}) = i\}$ and $p_i = \frac{|X_i|}{\sum_{j=1}^{K} |X_j|}$ for $i = 1, 2, \cdots, K$. Under the assumption that $\mathbf{c}_i$ is the arithmetic mean of $X_i$, we have

$$\mathrm{ReEff}_{\mathrm{mix}}^{\mathrm{SGM}} = \mathrm{ReEff}^{\mathrm{SGM}} - \sum_{i=1}^{K} p_i \|\mathbf{c}_i\|^2. \tag{21}$$

Proposition 2 provides a quantitative measurement of efforts reduction brought by the mixSGM, compared to the classical SGM. The amount of effort reduction reflects the amount of information provided by the structured prior distribution. One insight shown by Proposition 2 is that the effect of the effort reduction depends on $\sigma_T$, the standard deviation of the prior distribution in SGM. When $\sigma_T$ is very large, the impact of the reduction term $\sum_{i=1}^{K} p_i \|\mathbf{c}_i\|^2$ is minimal because both of the reverse efforts for the SGM and the mixSGM become significantly large. On the contrary, when $\sigma_T$ is moderate, the reduction effect becomes evident.

## 5 NUMERICAL EXPERIMENTS

### 5.1 OAKLAND CALL CENTER & PUBLIC WORK SERVICE REQUESTS DATASET

The Oakland Call Center & Public Works Service Requests Dataset is an open-source dataset containing service requests received by the Oakland Call Center. We preprocess the dataset to obtain the number of daily calls from July 1, 2009, to December 5, 2019. To learn the distribution of daily calls, we extract the number of daily calls from the 1,000th to the 2,279th day (a total of 1,280 days) since July 5, 2009, as the training data, and set the number of daily calls from the 2,280th to the 2,919th day (a total of 640 days) as the testing data. Since operational datasets often exhibit non-stationarity in terms of varying means, variances, and increasing (or decreasing) trends, we first conduct linear regression on the training data to eliminate potential trends and then normalize the data. We compare the effects of DDPM with mixDDPM. As the training data is one-dimensional, we utilize fully connected neural networks with the same architecture and an equal number of neurons. We train both models for 8k steps and independently generate 640 samples.

In Figure 3, we plot the density of the training data and the data generated by both DDPM and mixDDPM. We also calculate the $\mathcal{W}_1$ distances and the K-S statistics between the generated samples and the testing data, as shown in Table 4. The benchmark column is calculated by comparing the training data to the generated data, serving as a measurement of the distributional distances between the training and testing data. Relative errors are calculated as the difference between the metric values of the benchmark and the models, expressed as a fraction of the benchmark's metric value.

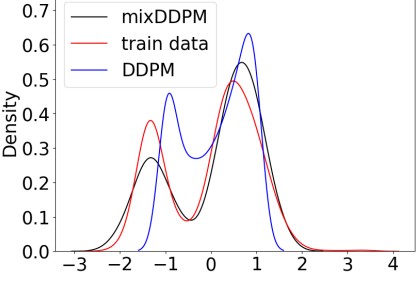

Table 4: DDPM v.s. mixDDPM on Oakland Call Center Datasets

|  | Benchmark | DDPM | mixDDPM |
|---|---|---|---|
| $\mathcal{W}_1$ Distance | 0.172 | 0.374 | 0.170 |
| $\mathcal{W}_1$ relative error |  | 1.174 | -0.012 |
| K-S statistics | 0.112 | 0.277 | 0.105 |
| K-S relative error |  | 1.473 | -0.063 |

Figure 3: DDPM and mixDDPM on Oakland Call Center Dataset

### 5.2 EXPERIMENTS ON EMNIST

In this section, we compare mixDDPM with DDPM using the EMNIST dataset Cohen et al. (2017), an extended version of MNIST that includes handwritten digits and characters in the format of $1 \times 28 \times 28$. We extract the first $N$ images of digits 0, 1, 2, and 3 to form the training dataset,

with $N$ values set to 64, 128, and 256. We select U-Net as the model architecture to learn the noise during training. As an illustrative example, we present the generated samples for $N = 128$ in Figure 4 below.

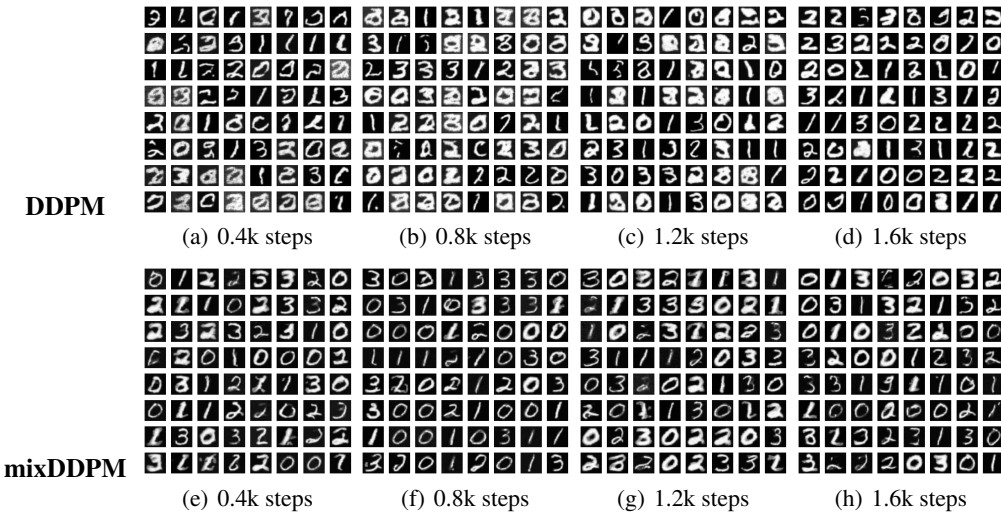

DDPM

      (a) 0.4k steps      (b) 0.8k steps      (c) 1.2k steps      (d) 1.6k steps

mixDDPM

      (e) 0.4k steps      (f) 0.8k steps      (g) 1.2k steps      (h) 1.6k steps

Figure 4: EMNIST Experiments with N=128

When training resources are limited, i.e., the number of training steps is relatively small, mixDDPM performs better than DDPM. Specifically, when the training step count is 0.4k, approximately one-quarter of the images generated by DDPM are difficult to identify visually, whereas only 10% of the images generated by mixDDPM are hard to identify. As the training step count increases to 1.6k, the sample quality of both DDPM and mixDDPM becomes visually comparable. This observation suggests that mixDDPM significantly improves the visual quality of the samples compared to DDPM when training resources are constrained. More experimental results, including variations in the size of the training data and the number of training steps, can be found in Appendix E.3. In addition, experiments for SGM and mixSGM can be found in Appendix E.3.

## 5.3 EXPERIMENTS ON CIFAR10

We test our model on CIFAR10, a dataset consisting of images with dimensions of $3 \times 32 \times 32$. We extract the first 2,560 images from three categories: dog, cat, and truck. These 7,680 images are fixed as the training data. During training, we use the same model architecture and noise schedule for both DDPM vs. mixDDPM and SGM vs. mixSGM to minimize the influence of other variables. We present the Fréchet Inception Distance (FID) Heusel et al. (2017) for the generated samples and the improvement ratio (Impr. Ratio) in Table 5 and Table 6. The improvement ratio is calculated as the difference between the FID for DDPM/SGM and the FID for mixDDPM/mixSGM, expressed as a fraction of the FID for DDPM/SGM. Additionally, we provide a comparison of generated samples in Figure 5.

Table 5: DDPM v.s. mixDDPM on CIFAR10

| Model \Training Steps | 180k | 240k | 300k | 360k | 420k | 480k | 540k | 600k |
|---|---|---|---|---|---|---|---|---|
| DDPM | 71.97 | 49.11 | 44.52 | 38.30 | 41.34 | 34.83 | 28.61 | 33.04 |
| mixDDPM | 35.84 | 23.43 | 20.78 | 18.15 | 16.43 | 13.82 | 14.80 | 12.88 |
| Impr. Ratio | 0.50 | 0.52 | 0.53 | 0.47 | 0.60 | 0.60 | 0.48 | 0.61 |

Table 6: SGM v.s. mixSGM on CIFAR10

| Model \Training Steps | 180k | 240k | 300k | 360k | 420k | 480k | 540k | 600k |
|---|---|---|---|---|---|---|---|---|
| SGM | 65.82 | 45.55 | 49.94 | 35.22 | 34.88 | 24.58 | 28.42 | 20.46 |
| mixSGM | 62.41 | 40.52 | 36.38 | 22.66 | 24.25 | 16.93 | 17.81 | 21.89 |
| Impr. Ratio | 0.05 | 0.11 | 0.27 | 0.36 | 0.30 | 0.31 | 0.37 | -0.07 |

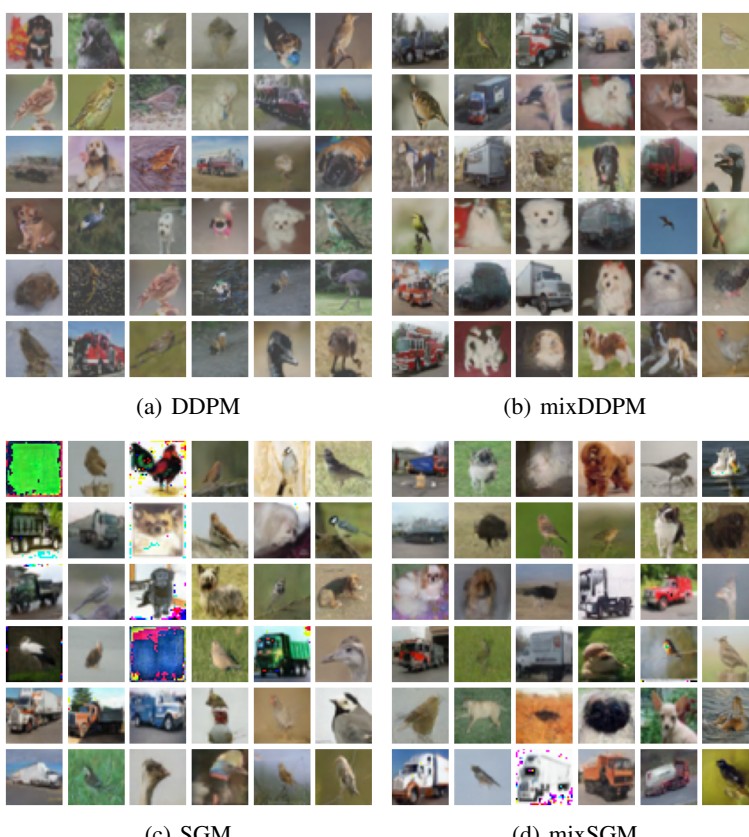

(a) DDPM                    (b) mixDDPM

(c) SGM                     (d) mixSGM

Figure 5: Experiments on CIFAR10 with 480k Training Steps

The results on CIFAR10 demonstrate that mixed diffusion models with Gaussian mixture priors generally achieve smaller FID scores (approximately 60% lower for mixDDPM and 3% lower for mixSGM) and better sample quality. The reduced FID and improved sample quality can be attributed to the utilization of the data distribution. By identifying suitable centers for the data distribution, the reverse process can begin from these centers instead of the zero point, thereby reducing the effort required during the reverse process. This leads to the improvements observed in the numerical results. Further implementation details and additional experimental results are provided in Appendix C.4 and E.4, respectively.

## 6 CONCLUSION

In this work, we propose and theoretically analyze a class of mixed diffusion models, where the prior distribution is chosen as mixed Gaussian distribution. The goal is to allow users to flexibly incorporate structured information or domain knowledge of the data into the prior distribution. The proposed model is shown to have advantageous comparative performance particularly when the training resources are limited. For future work, we plan to further the theoretical analysis and examine the performance of mixed diffusion models with data of different modalities.

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

# A    MIXSGM WITH VARIANCE ESTIMATION

As discussed in Section 4.2, the variance of each Gaussian component in the prior distribution of the mixSGM can be any arbitrary positive value, denoted by $\sigma_T^2$, and is not necessarily constrained to 1. In this section, we incorporate data-driven variance estimation for each component and provide numerical results to demonstrate the improvements achieved through variance estimation.

Given the number of components $K$, the parametric estimation of the prior distribution can be formalized as

$$\min_{\mathbf{c}_i,\sigma_i} \text{ReEff}_{\text{mix+var}}^{\text{SGM}} := \mathbb{E}_{\tilde{\mathbf{x}}_T \sim \mathcal{N}(\mathbf{c}_{D(\mathbf{x}_0)}, \sigma_{D(\mathbf{x}_0)}^2 \boldsymbol{I})} \mathbb{E}_{\mathbf{x}_0 \sim \bar{p}_{\text{data}}} \left[ \|\mathbf{x}_0 - \tilde{\mathbf{x}}_T\|^2 \right] . \tag{22}$$

Classical methods including Expectation Maximization algorithm Dempster et al. (1977) can be applied to solve the optimization problem (22). In addition, we provide a simpler method to estimate the variances based on the dispatcher $D$. To be more specific, we define

$$\hat{\sigma_i}^2 = \frac{1}{|\{\mathbf{x} : D(\mathbf{x}) = i\}|} \sum_{D(\mathbf{x}) = i} \frac{1}{d} \|\mathbf{x} - \mathbf{c}_i\|_2^2 \text{ for } i = 1, 2, \cdots, d, \tag{23}$$

where $d$ is the dimension of the state space.

With the given variance estimations $\sigma_i$, the forward SDE for the model starting from data samples $\mathbf{x}_0$ is given by

$$d\mathbf{x}_t = f_t(\mathbf{x}_t - \mathbf{c}_j)dt + \sigma_j g_t d\mathbf{w}_t, \ j = D(\mathbf{x}_0). \tag{24}$$

Following the notations in Section 4.2, the training procedure is to solve the optimization problem:

$$\min_{\theta} \mathcal{L}_{\text{mix+var}}^{\text{SGM}} = \int_0^T \omega_t \cdot \mathbb{E}_{\mathbf{x}_0, \epsilon} \left[ \|\epsilon_\theta(\mathbf{x}_t, t, j) - \epsilon\|_2^2 \right] dt, \tag{25}$$

where $\mathbf{x}_t = \alpha_t \mathbf{x}_0 + \mathbf{c}_{D(\mathbf{x}_0)} + \sigma_t \sigma_{D(\mathbf{x}_0)} \epsilon$. The current prior distribution can be written as $\sum_{i=1}^K p_i \mathcal{N}(\mathbf{c}_i, \sigma_i^2 \boldsymbol{I})$, where $p_i$ is the proportion of data that are assigned to the $i$-th center, as defined in Section 4. Moreover, the reverse SDE is given by

$$d\tilde{\mathbf{x}}_t = \left( f_t(\tilde{\mathbf{x}}_t - \mathbf{c}_j) + \frac{g_t^2 \sigma_j^2}{\sigma_t} \epsilon_\theta(\tilde{\mathbf{x}}_t, t, j) \right) dt + g_t \sigma_j d\tilde{\mathbf{w}}_t \tag{26}$$

given $\tilde{\mathbf{x}}_T$ comes from the $j$-th component, i.e, $\tilde{\mathbf{x}}_T \sim \mathcal{N}(\mathbf{c}_j, \sigma_j^2 \boldsymbol{I})$. For ease of exposition, we abbreviate the above mixSGM with variance estimation to mixSGM+var. Following the same experiment settings in Section 5.3, we compare FID score among the SGM, the mixSGM and the mixSGM+var in Table 7 below. All the Improvement Ratio (Impr. Ratio) are calculated with respect to the SGM.

Table 7: Experiment Result for mixSGM+var on CIFAR10

| Model \ Training Steps | 180k | 240k | 300k | 360k | 420k | 480k | 540k | 600k |
|---|---|---|---|---|---|---|---|---|
| SGM | 65.82 | 45.55 | 49.94 | 35.22 | 34.88 | 24.58 | 28.42 | 20.46 |
| mixSGM | 62.41 | 40.52 | 36.38 | 22.66 | 24.25 | 16.93 | 17.81 | 21.89 |
| Impr. Ratio | 0.05 | 0.11 | 0.27 | 0.36 | 0.30 | 0.31 | 0.37 | -0.07 |
| mixSGM+var | 51.22 | 36.17 | 29.58 | 22.17 | 18.05 | 16.65 | 15.73 | 13.09 |
| Impr. Ratio | 0.22 | 0.21 | 0.41 | 0.37 | 0.48 | 0.32 | 0.45 | 0.36 |

The results in Table 7 indicate that mixSGM+var consistently achieves lower FID scores compared to mixSGM. This finding further demonstrates the efficacy of the mixed diffusion model for image generation tasks, as the variance estimation method proposed in (23) requires minimal computation even in high-dimensional state spaces.

# B    VISUALIZATION OF 2D EXAMPLE

In this section, we take 2D diffusion model as an illustrative example to show how smaller reverse effort help improve the sample quality. Throughout the section, we choose the data distribution to

be

$$\frac{1}{4}\Big(\mathcal{N}((1,1),0.01\boldsymbol{I}) + \mathcal{N}((-1,1),0.01\boldsymbol{I}) + \mathcal{N}((1,-1),0.01\boldsymbol{I}) + \mathcal{N}((-1,-1),0.01\boldsymbol{I})\Big), \quad (27)$$

which is a 2D Gaussian mixture distribution.

We draw 5,120 samples from this distribution and use the same data samples to independently train both the DDPM and the mixDDPM, using the same noise schedule and time horizon. Additionally, the depth of the two neural networks and the width of each layer are kept the same. During the reverse sampling process, we draw 2560 samples from each model. We plot the sample paths of the reverse process for both models in Figure 6 below. The blue points represent samples from the reverse process, drawn at six equidistant time points: 1000, 800, 600, 400, 200 and 0. We also apply the Silhouette method Rousseeuw (1987) to determine the optimal number of clusters from the reverse process samples and plot the cluster centers with red points.

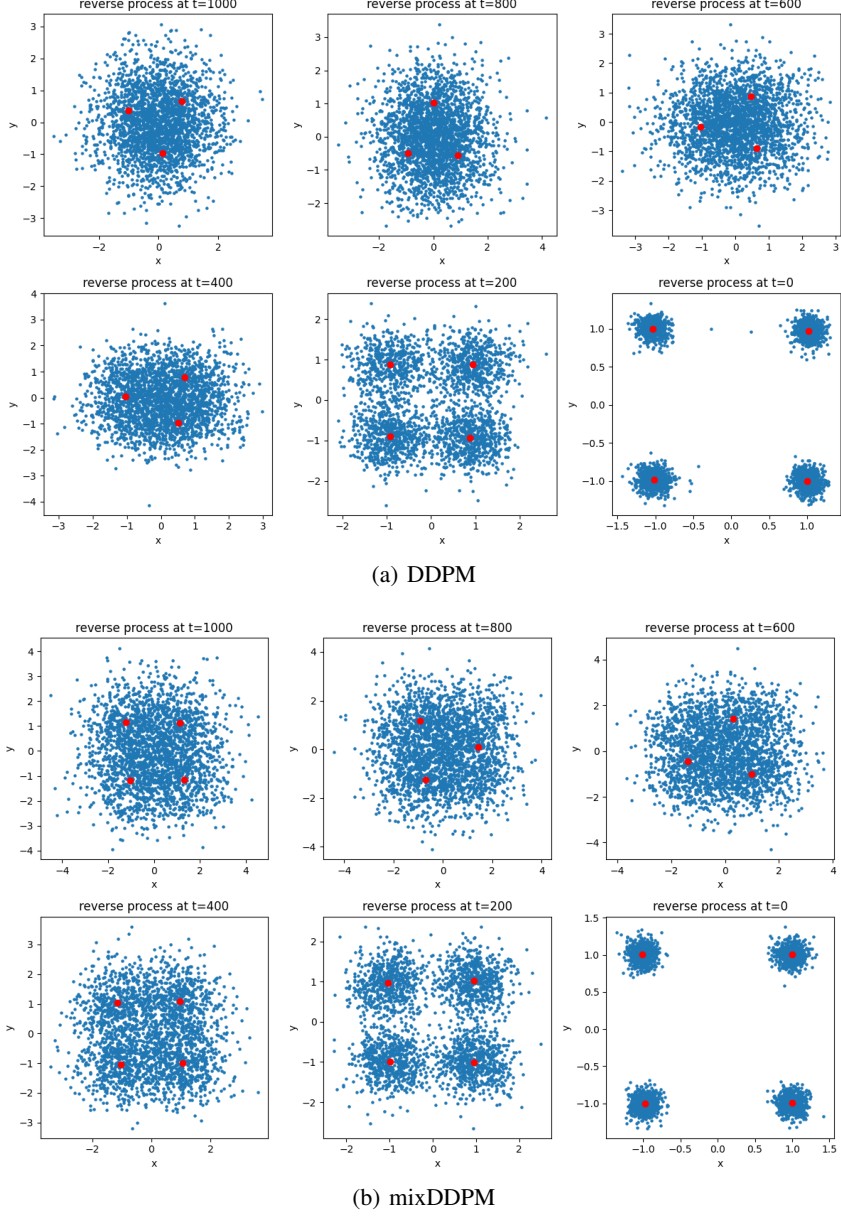

Figure 6: The reverse process

The visualizations of the reverse process reveal that mixDDPM forms four clusters at $t = 400$, while DDPM samples still fail to exhibit clear clustering at this stage. By $t = 200$, mixDDPM produces well-defined clusters, whereas DDPM exhibits more points distributed between nearby clusters. Ultimately, mixDDPM demonstrates better sampling quality since the variance of each cluster is closer to the data distribution, compared to samples generated by DDPM. These findings indicate that mixDDPM, by reducing the reverse effort, alleviates the model's burden and facilitates more efficient sampling by producing clusters earlier and more distinctly than DDPM.

## C  ADDITIONAL IMPLEMENTATION DETAILS

### C.1  CENTER SELECTION METHODS

In this section, we provide several center selection methods based on the analysis of the training data. As mentioned in Section 4, the specific method can be determined by users and does not need to be optimal.

1. **Data-driven clustering method.** The data-driven clustering method first applies a traditional data-clustering technique Rousseeuw (1987); Kaufman & Rousseeuw (2009) to the samples from the data distribution. To be more specific, the method first calculates the average Silhouette's coefficient for samples from the data distribution under different number of clusters. By maximizing the average Silhouette's coefficient over different values of $K$, the method determines an optimal value of $K$. Subsequently, the method applies the k-means algorithm to find the $K$ cluster centers for the data distribution and we denote them by $\mathbf{c}_1, \cdots, \mathbf{c}_K \in \mathbb{R}^d$. We summarize the implementation of this method in Algorithm 3 below.

---

**Algorithm 3** Data-driven Clustering Method

---

**Input:** Datasets $\mathcal{S}$, maximum $K$ value $K_{\max}$
**Output:** the number of clusters $K$ and the centers $\mathbf{c}_1, \cdots, \mathbf{c}_K$
$SC \leftarrow [0] * (K_{\max} - 1)$
**for** $k = 2$ to $K_{\max}$ **do**
  Apply the k-means algorithm to find $k$ cluster centers $\mathbf{c}_1^k, \cdots, \mathbf{c}_k^k$
  **for** $\mathbf{x}_0$ in $\mathcal{S}$ **do**
    $SC[k-2] + =$ Silhouette's coefficient of $\mathbf{x}_0$
  **end for**
  $K' = \arg\min_k SC[k]$
**end for**
**Return** $K = K' + 2$ and $\mathbf{c}_1^K, \cdots, \mathbf{c}_K^K$

---

2. **Data Labeling.** When samples from the data distribution have either pre-given labels or can be labeled through pre-trained classifier, the labels naturally separate the samples into several groups. Hence, the mixed diffusion models can follow the number of different labels and the centers among samples with the same label to determine the value of $K$ and $\mathbf{c}_1, \cdots, \mathbf{c}_K$.

3. **Alternative methods.** Alternatively, the number of clusters $K$ and the centers of the clusters $\mathbf{c}_1, \cdots, \mathbf{c}_K \in \mathbb{R}^d$ can be seen as pre-given hyperparameters that are possibly specified by domain knowledge or other preliminary data analysis.

### C.2  ALGORITHMS FOR THE MIXSGM

Algorithms for the training and sampling process of the mixSGM are shown in Algorithm 4 and 5 below.

---

**Algorithm 4** Training Process for the Mixed SGM

---

**Input:** samples $\mathbf{x}_0$ from the data distribution, un-trained neural network $\epsilon_\theta$, time horizon $T$, scalar function $f, g$, number of centers $K$ and the centers $\mathbf{c}_1, \cdots, \mathbf{c}_K$
**Output:** Trained neural network $\epsilon_\theta$
Calculate $\alpha_t$ and $\sigma_t$ in closed-form
**repeat**
    Get data $\mathbf{x}_0$
    Find center $j = D(\mathbf{x}_0)$
    Sample $t \sim U[0, T]$ and $\epsilon \sim \mathcal{N}(\mathbf{0}, \boldsymbol{I})$
    $\mathbf{x}_t \leftarrow \alpha_t \mathbf{x}_0 + \mathbf{c}_j + \sigma_t \epsilon$
    $\mathcal{L} \leftarrow \omega_t \|\epsilon - \epsilon_\theta(\mathbf{x}_t, t, j)\|_2^2$
    Take a gradient descent step on $\nabla_\theta \mathcal{L}$
**until** Converged or training resource/time limit is hit

---

**Algorithm 5** Reverse Process for the Mixed SGM

---

**Input:** Trained neural network $\epsilon_\theta$, center weights $p_1, \cdots, p_K$, centers $\mathbf{c}_1, \cdots, \mathbf{c}_K$
Sample $j \in \{1, \cdots, K\}$ with $\mathbb{P}(j = i) = p_i$ for $i = 1, \cdots, K$
Sample $\tilde{\mathbf{x}}_T \sim \mathcal{N}(\mathbf{c}_j, \sigma_T^2 \boldsymbol{I})$
Apply numerical solvers to the reverse SDE (18).
**Return** $\tilde{\mathbf{x}}_0$

---

### C.3 IMPLEMENTATION DETAILS ON EMNIST

We apply the U-Net architecture to learn the noise during the training of both the original diffusion models and the mixed diffusion models. The down-sampling path consists of three blocks with progressively increasing output channels. The specific number of output channels are 32, 64, and 128. The third block incorporates attention mechanisms to capture global context. Similarly, the up-sampling path mirrors the down-sampling structure, with the first block replaced by an attention block to refine spatial details. For the mixed diffusion models, we use class embeddings to incorporate the assignments from the dispatcher and employ a data-driven clustering method, as described in Algorithm 3. The data are preprocessed with a batch size of 16.

For both DDPM and mixDDPM, we set the time step to $T = 1000$ and choose the noise schedule $\beta_t$ as a linear function of $t$, with $\beta_1 = 0.001$ and $\beta_{1000} = 0.02$. For the SGM, we select the following forward SDE:

$$d\mathbf{x}_t = -\frac{1}{2}\beta_t \mathbf{x}_t dt + \sqrt{\beta_t} d\mathbf{w}_t. \tag{28}$$

For mixSGM, the forward SDE is defined as:

$$d\mathbf{x}_t = -\frac{1}{2}\beta_t (\mathbf{x}_t - \mathbf{c}_j) dt + \sqrt{\beta_t} d\mathbf{w}_t, \quad \text{where } j = D(\mathbf{x}_0). \tag{29}$$

Here, $\beta_t$ is chosen to be a linear function with $\beta_0 = 0.1$ and $\beta_1 = 40$ for both SGM and mixSGM. We use the DPM solver Lu et al. (2022a) for efficient sampling.

### C.4 IMPLEMENTATION DETAILS ON CIFAR10

For both the original and the mixed diffusion models, we apply the U-Net architecture, which consists of a series of down-sampling and up-sampling blocks, with each block containing two layers. The down-sampling path has five blocks with progressively increasing output channels. The specific number of output channels are 32, 64, 128, 256, and 512. Among these, the fourth block integrates attention mechanisms to capture global context. Similarly, the up-sampling path mirrors the down-sampling structure, with the second block replaced by an attention block to refine spatial details. A dropout rate of 0.1 is applied to regularize the model. Specifically for the mixed diffusion models, the model utilizes class embeddings to incorporate the assignment provided by the dispatcher. Before training the neural networks, we first scale the training data to the range of $[-2, 2]$ with a batch size of 128. We choose the weighting function $\omega_t$ in (11) to be 1, regardless of the time step.

Since the images in CIFAR10 are already labeled, we adopt a data labeling method to determine the number of centers $K$ and the centers $\mathbf{c}_1, \cdots, \mathbf{c}_K$.

For DDPM and mixDDPM, the noise schedules are set with $\beta_1 = 0.001$ and $\beta_{1000} = 0.02$, following a linear schedule over 1000 steps. For SGM, we choose the forward SDE as:

$$d\mathbf{x}_t = -\frac{1}{2}\beta_t \mathbf{x}_t dt + \sqrt{\beta_t} d\mathbf{w}_t. \tag{30}$$

For mixSGM, we choose the forward SDE as:

$$d\mathbf{x}_t = -\frac{1}{2}\beta_t(\mathbf{x}_t - \mathbf{c}_j)dt + \sqrt{\beta_t} d\mathbf{w}_t, \text{ where } j = D(\mathbf{x}_0). \tag{31}$$

Here, $\beta_t$ is chosen as a linear function with $\beta_0 = 0.1$ and $\beta_1 = 40$ for both SGM and mixSGM. We set the batch size to 128 and apply the DPM solver Lu et al. (2022a) for efficient sampling.

## D  PROOF

***Proof of Proposition 1.*** We first calculate $\text{Eff}^{\text{DDPM}}$. Since $\mathbf{x}_0$ and $\tilde{\mathbf{x}}_T$ are independent, we have

$$\begin{aligned}
\text{Eff}^{\text{DDPM}} &= \mathbb{E}_{\mathbf{x}_0 \sim \bar{p}_{\text{data}}}\left[\|\mathbf{x}_0\|^2\right] + \mathbb{E}_{\tilde{\mathbf{x}}_T \sim \mathcal{N}(\mathbf{0}, \boldsymbol{I})}\left[\|\tilde{\mathbf{x}}_T\|^2\right] - 2\mathbb{E}_{\tilde{\mathbf{x}}_T \sim \mathcal{N}(\mathbf{0}, \boldsymbol{I}),\, \mathbf{x}_0 \sim \bar{p}_{\text{data}}}\left[\mathbf{x}_0^T \tilde{\mathbf{x}}_T\right] \\
&= \mathbb{E}_{\mathbf{x}_0 \sim \bar{p}_{\text{data}}}\left[\|\mathbf{x}_0\|^2\right] + d,
\end{aligned} \tag{32}$$

where $d$ is the dimension of the state space. On the contrary, we calculate $\text{Eff}_{\text{mix}}^{\text{DDPM}}$ by first conditioning on the centers:

$$\begin{aligned}
\text{Eff}_{\text{mix}}^{\text{DDPM}} &= \mathbb{E}\left[\mathbb{E}_{\tilde{\mathbf{x}}_T \sim \mathcal{N}(\mathbf{c}_{D(\mathbf{x}_0)}, \boldsymbol{I})}\mathbb{E}_{\mathbf{x}_0 \sim \bar{p}_{\text{data}}}\left[\|\mathbf{x}_0 - \tilde{\mathbf{x}}_T\|^2\right]\Big| D(\mathbf{x}_0) = i\right] \\
&= \sum_{i=1}^{K} p_i \mathbb{E}_{\tilde{\mathbf{x}}_T \sim \mathcal{N}(\mathbf{c}_i, \boldsymbol{I}),\, \mathbf{x}_0 \sim \bar{p}_{\text{data}}|X_i}\left[\|\mathbf{x}_0 - \tilde{\mathbf{x}}_T\|^2\right] \\
&= \sum_{i=1}^{K} p_i \Big(\mathbb{E}_{\mathbf{x}_0 \sim \bar{p}_{\text{data}}|X_i}\left[\|\mathbf{x}_0\|^2\right] + \mathbb{E}_{\tilde{\mathbf{x}}_T \sim \mathcal{N}(\mathbf{c}_i, \boldsymbol{I})}\left[\|\tilde{\mathbf{x}}_T\|^2\right] \\
&\qquad - 2\mathbb{E}_{\tilde{\mathbf{x}}_T \sim \mathcal{N}(\mathbf{c}_i, \boldsymbol{I}),\, \mathbf{x}_0 \sim \bar{p}_{\text{data}}|X_i}\left[\mathbf{x}_0^T \tilde{\mathbf{x}}_T\right]\Big) \\
&= \mathbb{E}_{\mathbf{x}_0 \sim \bar{p}_{\text{data}}}\left[\|\mathbf{x}_0\|^2\right] + d + \sum_{i=1}^{K} p_i\|\mathbf{c}_i\|^2 - 2\sum_{i=1}^{K} p_i \mathbb{E}_{\tilde{\mathbf{x}}_T \sim \mathcal{N}(\mathbf{c}_i, \boldsymbol{I}),\, \mathbf{x}_0 \sim \bar{p}_{\text{data}}|X_i}\left[\mathbf{x}_0^T \tilde{\mathbf{x}}_T\right].
\end{aligned} \tag{33}$$

Since $\mathbf{x}_0$ and $\tilde{\mathbf{x}}_T$ are independent conditioned on the subspace $X_i$, we obtain

$$\mathbb{E}_{\tilde{\mathbf{x}}_T \sim \mathcal{N}(\mathbf{c}_i, \boldsymbol{I}),\, \mathbf{x}_0 \sim \bar{p}_{\text{data}}|X_i}\left[\mathbf{x}_0^T \tilde{\mathbf{x}}_T\right] = \mathbb{E}_{\mathbf{x}_0 \sim \bar{p}_{\text{data}}|X_i}\left[\mathbf{x}_0\right]^T \mathbb{E}_{\tilde{\mathbf{x}}_T \sim \mathcal{N}(\mathbf{c}_i, \boldsymbol{I})}\left[\tilde{\mathbf{x}}_T\right] = \mathbf{c}_i^T \mathbf{c}_i = \|\mathbf{c}_i\|^2. \tag{34}$$

Hence, the effort of the mixDDPM is given by

$$\text{Eff}_{\text{mix}}^{\text{DDPM}} = \mathbb{E}_{\mathbf{x}_0 \sim \bar{p}_{\text{data}}}\left[\|\mathbf{x}_0\|^2\right] + d - \sum_{i=1}^{K} p_i\|\mathbf{c}_i\|^2. \tag{35}$$

Combining (32) and (35), we finish the proof for (16). $\qquad\qquad\square$

***Proof of Proposition 2.*** To prove Proposition 2, we first calculate $\text{Eff}^{\text{SGM}}$. Since $\mathbf{x}_0$ and $\tilde{\mathbf{x}}_T$ are independent, we have

$$\begin{aligned}
\text{Eff}^{\text{SGM}} &= \mathbb{E}_{\mathbf{x}_0 \sim \bar{p}_{\text{data}}}\left[\|\mathbf{x}_0\|^2\right] + \mathbb{E}_{\tilde{\mathbf{x}}_T \sim \mathcal{N}(\mathbf{0}, \sigma_T^2 \boldsymbol{I})}\left[\|\tilde{\mathbf{x}}_T\|^2\right] - 2\mathbb{E}_{\tilde{\mathbf{x}}_T \sim \mathcal{N}(\mathbf{0}, \sigma_T^2 \boldsymbol{I}),\, \mathbf{x}_0 \sim \bar{p}_{\text{data}}}\left[\mathbf{x}_0^T \tilde{\mathbf{x}}_T\right] \\
&= \mathbb{E}_{\mathbf{x}_0 \sim \bar{p}_{\text{data}}}\left[\|\mathbf{x}_0\|^2\right] + \sigma_T^2 d,
\end{aligned} \tag{36}$$

where $d$ is the dimension of the state space. On the contrary, we calculate $\text{Eff}_{\text{mix}}^{\text{SGM}}$ by first conditioning on the centers:

$$
\begin{aligned}
\text{Eff}_{\text{mix}}^{\text{SGM}} &= \mathbb{E}\left[\mathbb{E}_{\tilde{\mathbf{x}}_T \sim \mathcal{N}(\mathbf{c}_{D(\mathbf{x}_0)}, \sigma_T^2 \mathbf{I})} \mathbb{E}_{\mathbf{x}_0 \sim \bar{p}_{\text{data}}}\left[\|\mathbf{x}_0 - \tilde{\mathbf{x}}_T\|^2\right]\Big|D(\mathbf{x}_0) = i\right] \\
&= \sum_{i=1}^{K} p_i \mathbb{E}_{\tilde{\mathbf{x}}_T \sim \mathcal{N}(\mathbf{c}_i, \sigma_T^2 \mathbf{I}),\ \mathbf{x}_0 \sim \bar{p}_{\text{data}}|_{X_i}}\left[\|\mathbf{x}_0 - \tilde{\mathbf{x}}_T\|^2\right] \\
&= \sum_{i=1}^{K} p_i \Big(\mathbb{E}_{\mathbf{x}_0 \sim \bar{p}_{\text{data}}|_{X_i}}\left[\|\mathbf{x}_0\|^2\right] + \mathbb{E}_{\tilde{\mathbf{x}}_T \sim \mathcal{N}(\mathbf{c}_i, \sigma_T^2 \mathbf{I})}\left[\|\tilde{\mathbf{x}}_T\|^2\right] \\
&\qquad - 2\mathbb{E}_{\tilde{\mathbf{x}}_T \sim \mathcal{N}(\mathbf{c}_i, \sigma_T^2 \mathbf{I}),\ \mathbf{x}_0 \sim \bar{p}_{\text{data}}|_{X_i}}\left[\mathbf{x}_0^T \tilde{\mathbf{x}}_T\right]\Big) \\
&= \mathbb{E}_{\mathbf{x}_0 \sim \bar{p}_{\text{data}}}\left[\|\mathbf{x}_0\|^2\right] + \sigma_T^2 d + \sum_{i=1}^{K} p_i \|\mathbf{c}_i\|^2 - 2\sum_{i=1}^{K} p_i \mathbb{E}_{\tilde{\mathbf{x}}_T \sim \mathcal{N}(\mathbf{c}_i, \mathbf{I}),\ \mathbf{x}_0 \sim \bar{p}_{\text{data}}|_{X_i}}\left[\mathbf{x}_0^T \tilde{\mathbf{x}}_T\right].
\end{aligned}
\tag{37}
$$

Since $\mathbf{x}_0$ and $\tilde{\mathbf{x}}_T$ are independent conditioned on the subspace $X_i$, we obtain

$$
\mathbb{E}_{\tilde{\mathbf{x}}_T \sim \mathcal{N}(\mathbf{c}_i, \mathbf{I}),\ \mathbf{x}_0 \sim \bar{p}_{\text{data}}|_{X_i}}\left[\mathbf{x}_0^T \tilde{\mathbf{x}}_T\right] = \mathbb{E}_{\mathbf{x}_0 \sim \bar{p}_{\text{data}}|_{X_i}}\left[\mathbf{x}_0\right]^T \mathbb{E}_{\tilde{\mathbf{x}}_T \sim \mathcal{N}(\mathbf{c}_i, \mathbf{I})}\left[\tilde{\mathbf{x}}_T\right] = \mathbf{c}_i^T \mathbf{c}_i = \|\mathbf{c}_i\|^2. \tag{38}
$$

Hence, the effort of the mixSGM is given by

$$
\text{Eff}_{\text{mix}}^{\text{SGM}} = \mathbb{E}_{\mathbf{x}_0 \sim \bar{p}_{\text{data}}}\left[\|\mathbf{x}_0\|^2\right] + \sigma_T^2 d - \sum_{i=1}^{K} p_i \|\mathbf{c}_i\|^2. \tag{39}
$$

Combining (36) and (39), we finish the proof for Proposition 2. $\qquad\square$

# E ADDITIONAL NUMERICAL RESULTS

## E.1 EXPERIMENT RESULTS ON CELEBA-HQ

In this section, we compare mixDDPM with DDPM on CelebA-HQ Karras (2017), a high-resolution facial image dataset. We extract a total of 12,800 images, evenly split between male and female subjects. We re-size the images to $128 \times 128$ resolution and choose $K = 2$ for mixDDPM.

For both the classic and the mixed diffusion models, we apply the U-Net architecture, which consists of a series of down-sampling and up-sampling blocks, with each block containing two layers. The down-sampling path has five blocks with progressively increasing output channels. The specific number of output channels are 64, 128, 128, 256 and 256. Among these, the fourth block integrates attention mechanisms to capture global context. Similarly, the up-sampling path mirrors the down-sampling structure, with the second block replaced by an attention block to refine spatial details. A dropout rate of 0.1 is applied to regularize the model.

Below are the generated samples from DDPM and mixDDPM with same experimental settings.

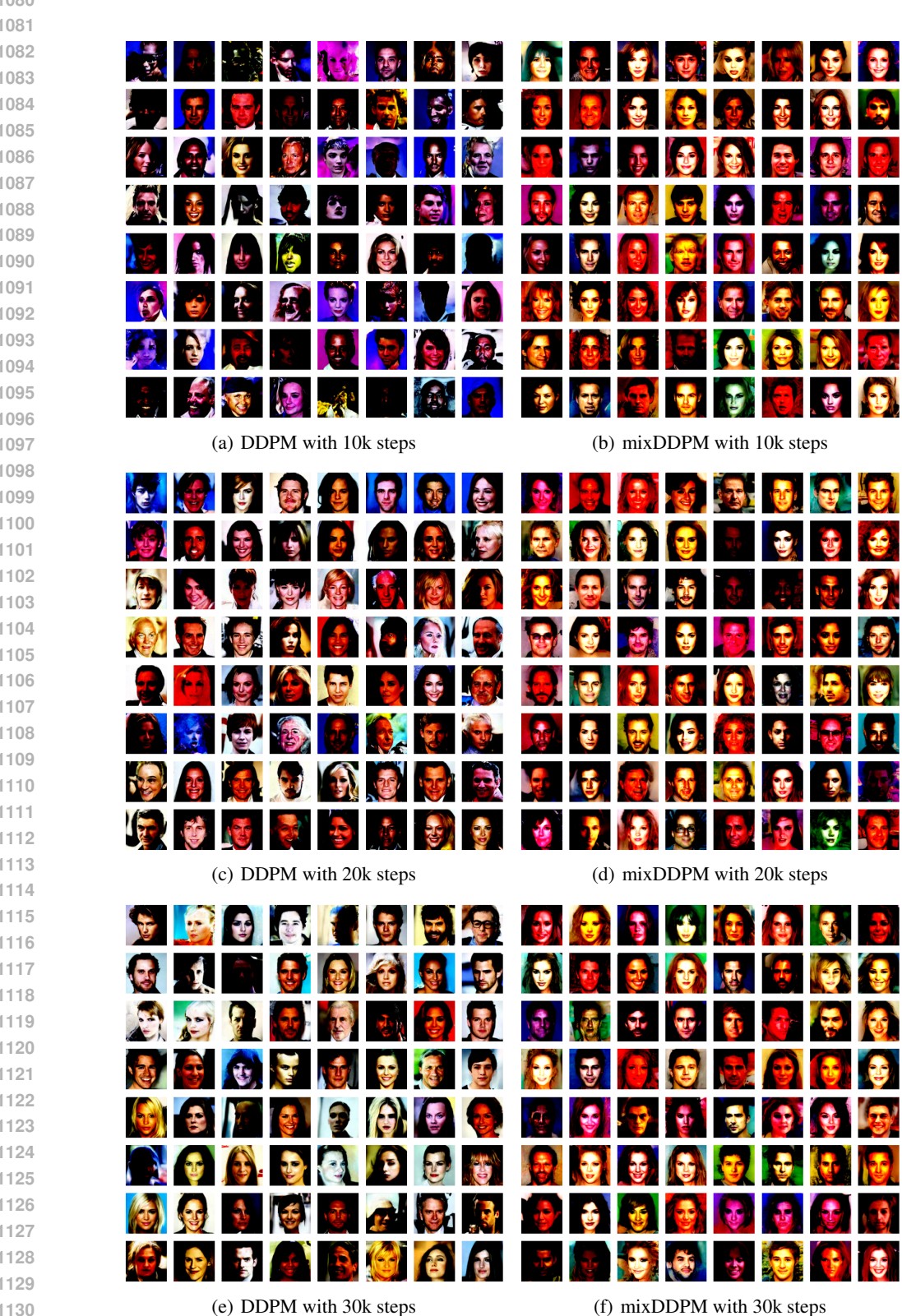

(a) DDPM with 10k steps                    (b) mixDDPM with 10k steps

(c) DDPM with 20k steps                    (d) mixDDPM with 20k steps

(e) DDPM with 30k steps                    (f) mixDDPM with 30k steps

Figure 7: Experiment results on CelebA-HQ

## E.2 ADDITIONAL EXPERIMENT RESULTS ON OAKLAND CALL CENTER DATASET

We present in this section the numerical results for SGM and mixSGM on the Oakland Call Center experiment. For this experiment, the training steps is 4k.

Table 8: SGM v.s. mixSGM on Oakland Call Center Datasets

|  | Benchmark | SGM | mixSGM |
|---|---|---|---|
| $\mathcal{W}_1$ Distance | 0.172 | 0.400 | 0.228 |
| $\mathcal{W}_1$ relative error |  | 1.325 | 0.326 |
| K-S statistics | 0.112 | 0.189 | 0.144 |
| K-S relative error |  | 0.688 | 0.286 |

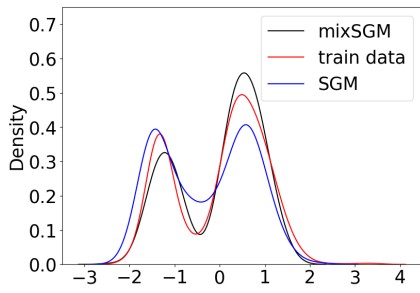

Figure 8: SGM and mixSGM on Oakland Call Center Dataset

## E.3 ADDITIONAL EXPERIMENT RESULTS ON EMNIST

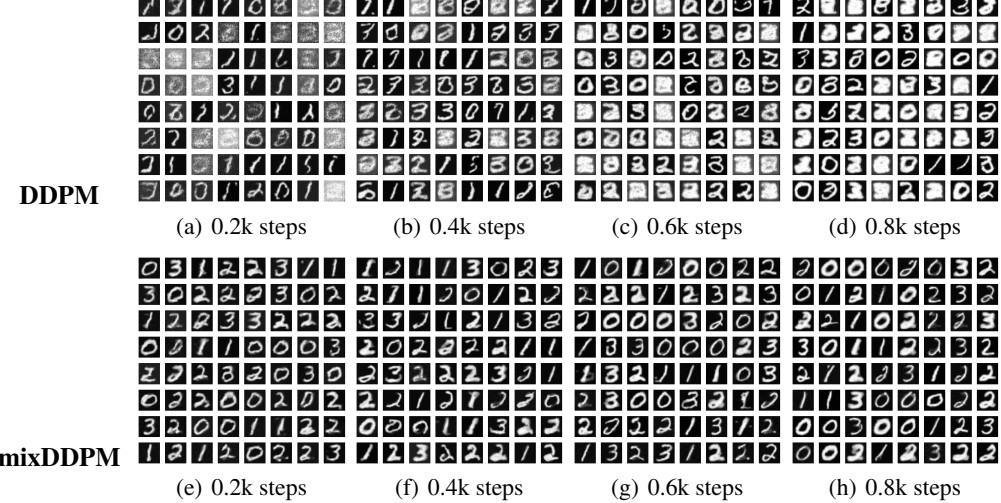

Figure 9: EMNIST Experiments with $N = 64$

**DDPM**

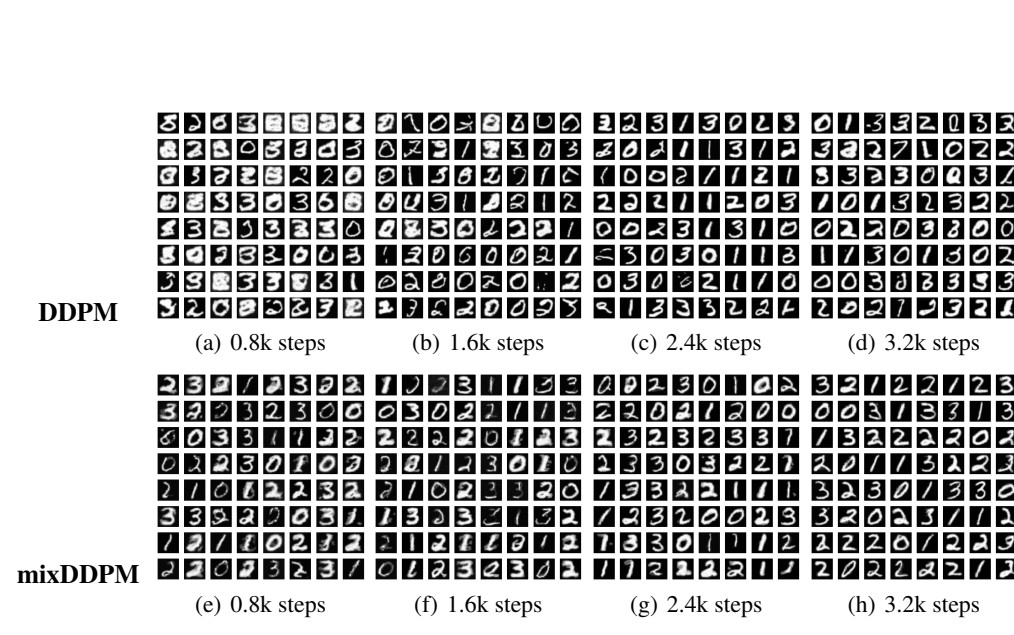

| (a) 0.8k steps | (b) 1.6k steps | (c) 2.4k steps | (d) 3.2k steps |

**mixDDPM**

| (e) 0.8k steps | (f) 1.6k steps | (g) 2.4k steps | (h) 3.2k steps |

Figure 10: EMNIST Experiments with $N = 256$

**SGM**

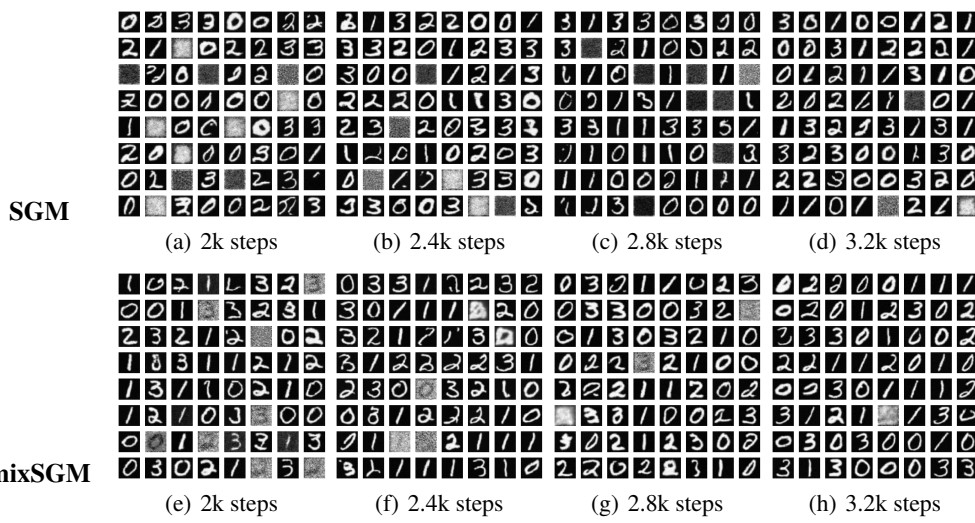

| (a) 2k steps | (b) 2.4k steps | (c) 2.8k steps | (d) 3.2k steps |

**mixSGM**

| (e) 2k steps | (f) 2.4k steps | (g) 2.8k steps | (h) 3.2k steps |

Figure 11: EMNIST Experiments with $N = 128$

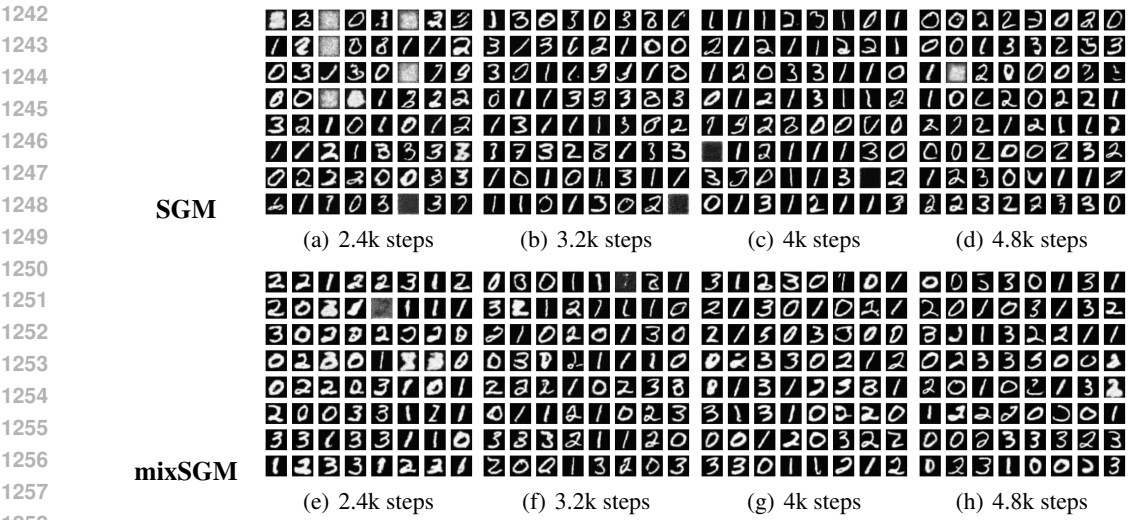

**SGM**

(a) 2.4k steps     (b) 3.2k steps     (c) 4k steps     (d) 4.8k steps

**mixSGM**

(e) 2.4k steps     (f) 3.2k steps     (g) 4k steps     (h) 4.8k steps

Figure 12: EMNIST Experiments with $N = 256$

## E.4 ADDITIONAL EXPERIMENT RESULTS ON CIFAR10

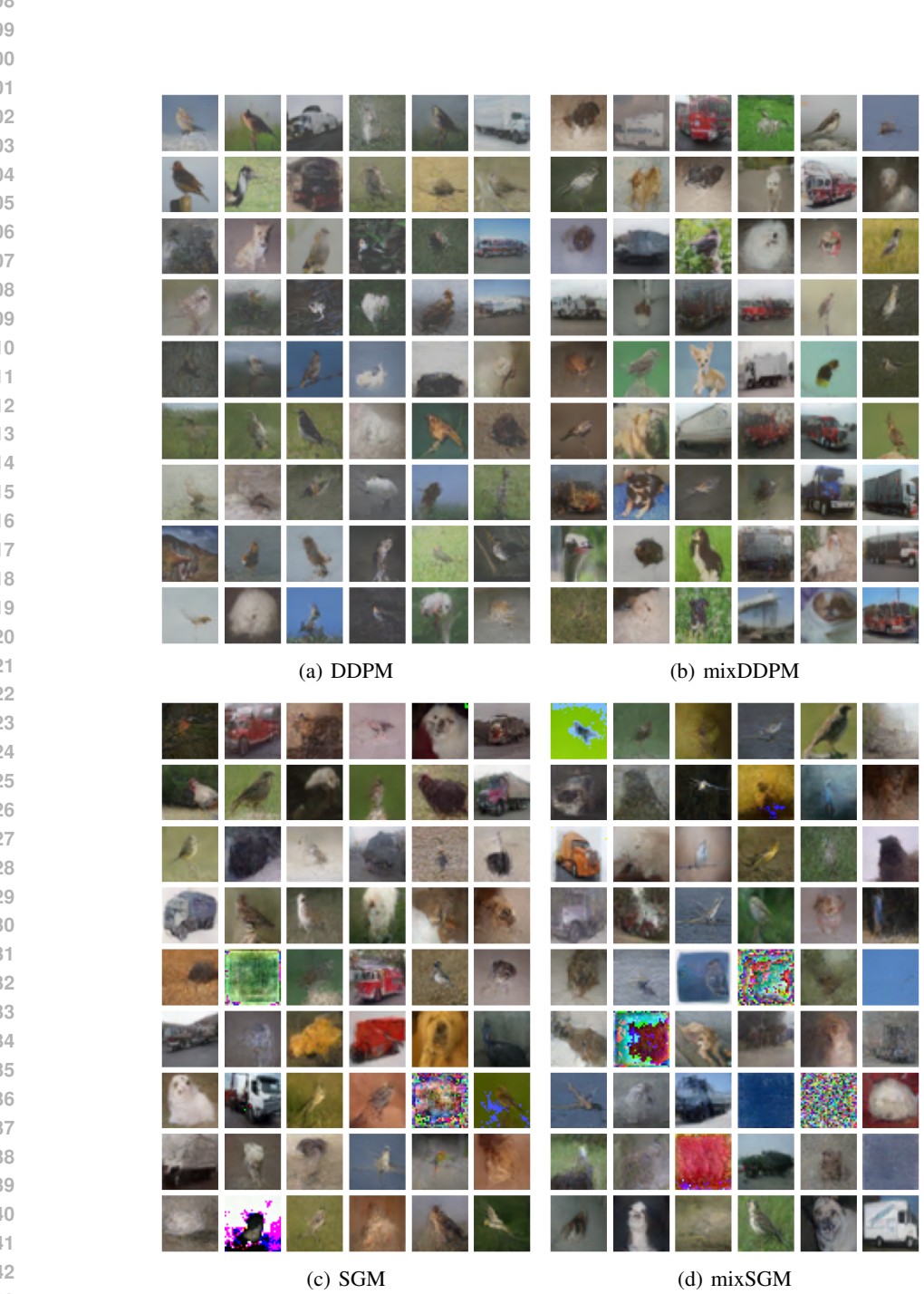

(a) DDPM (b) mixDDPM

(c) SGM (d) mixSGM

Figure 13: Experiments on CIFAR10 with 180k training steps

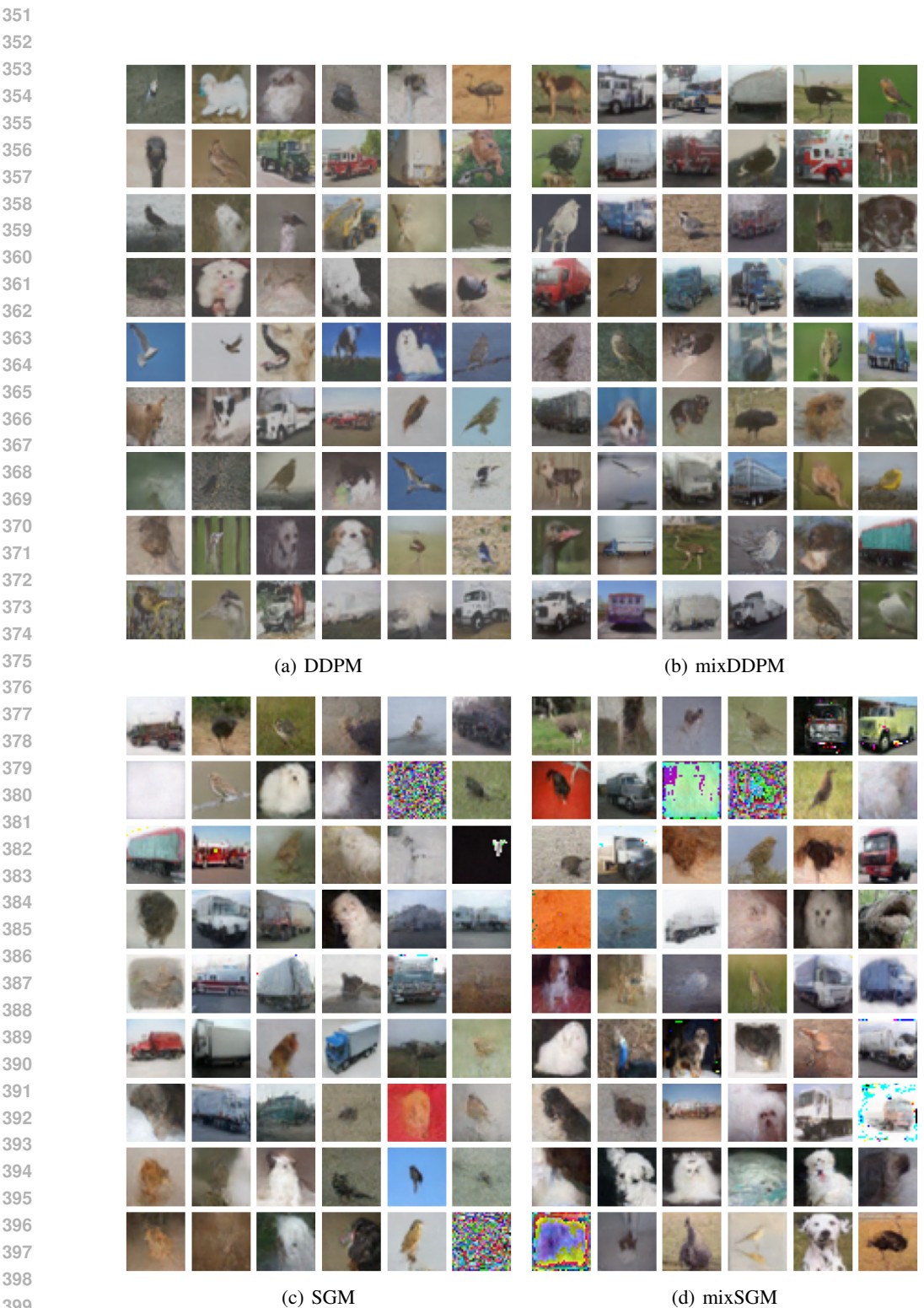

(a) DDPM  (b) mixDDPM

(c) SGM  (d) mixSGM

Figure 14: Experiments on CIFAR10 with 240k training steps

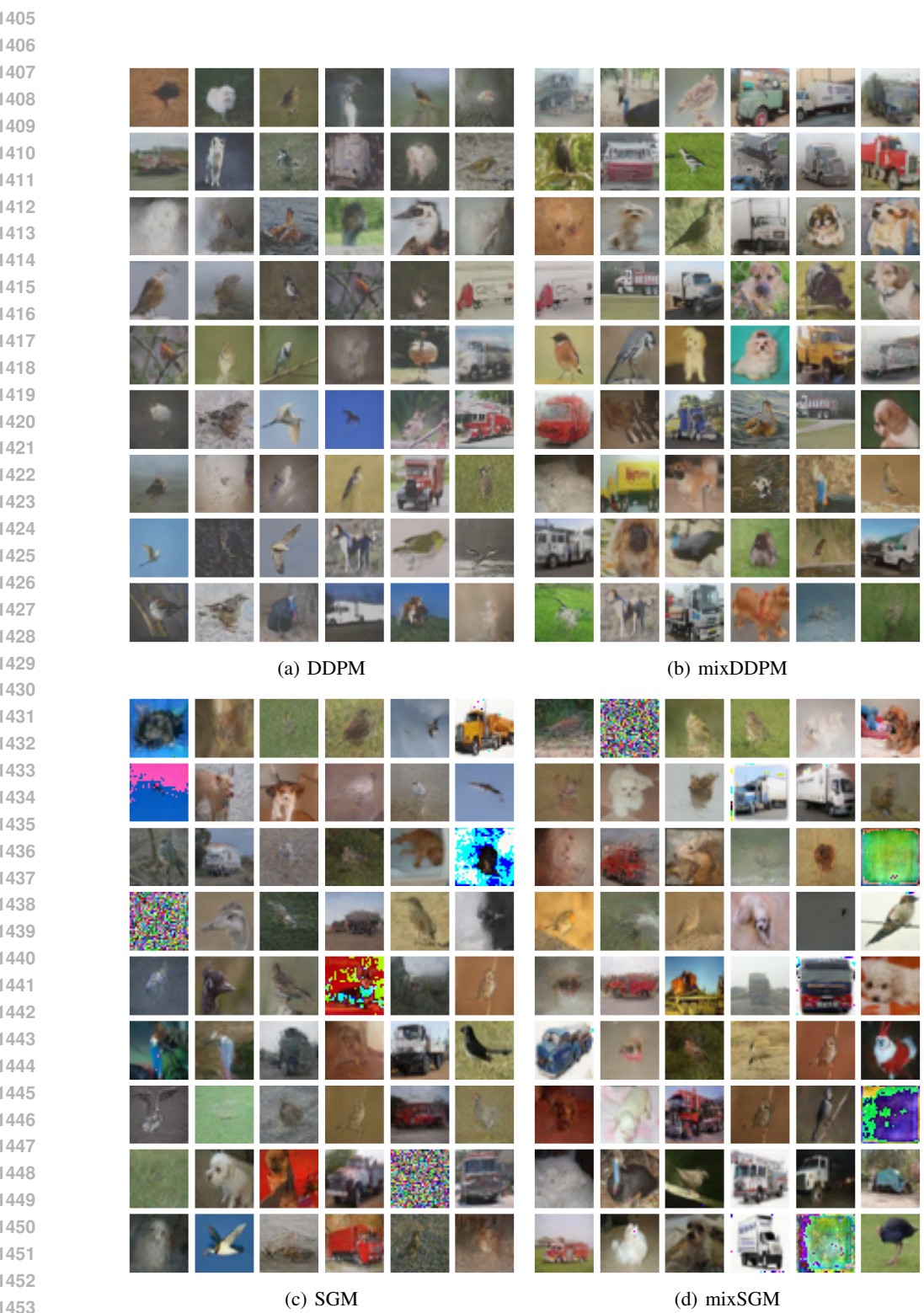

(a) DDPM

(b) mixDDPM

(c) SGM

(d) mixSGM

Figure 15: Experiments on CIFAR10 with 300k training steps

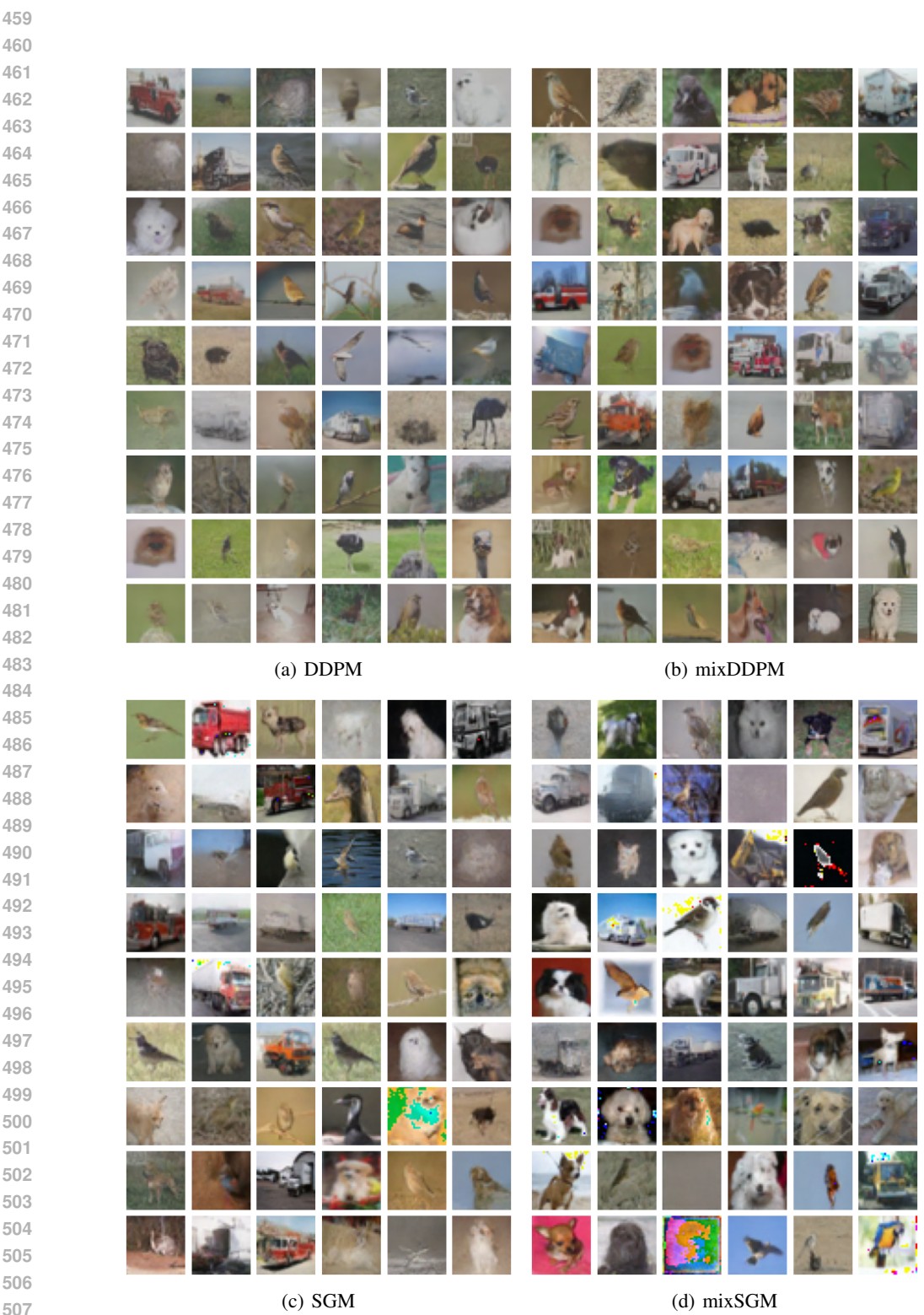

(a) DDPM          (b) mixDDPM

(c) SGM          (d) mixSGM

Figure 16: Experiments on CIFAR10 with 360k training steps

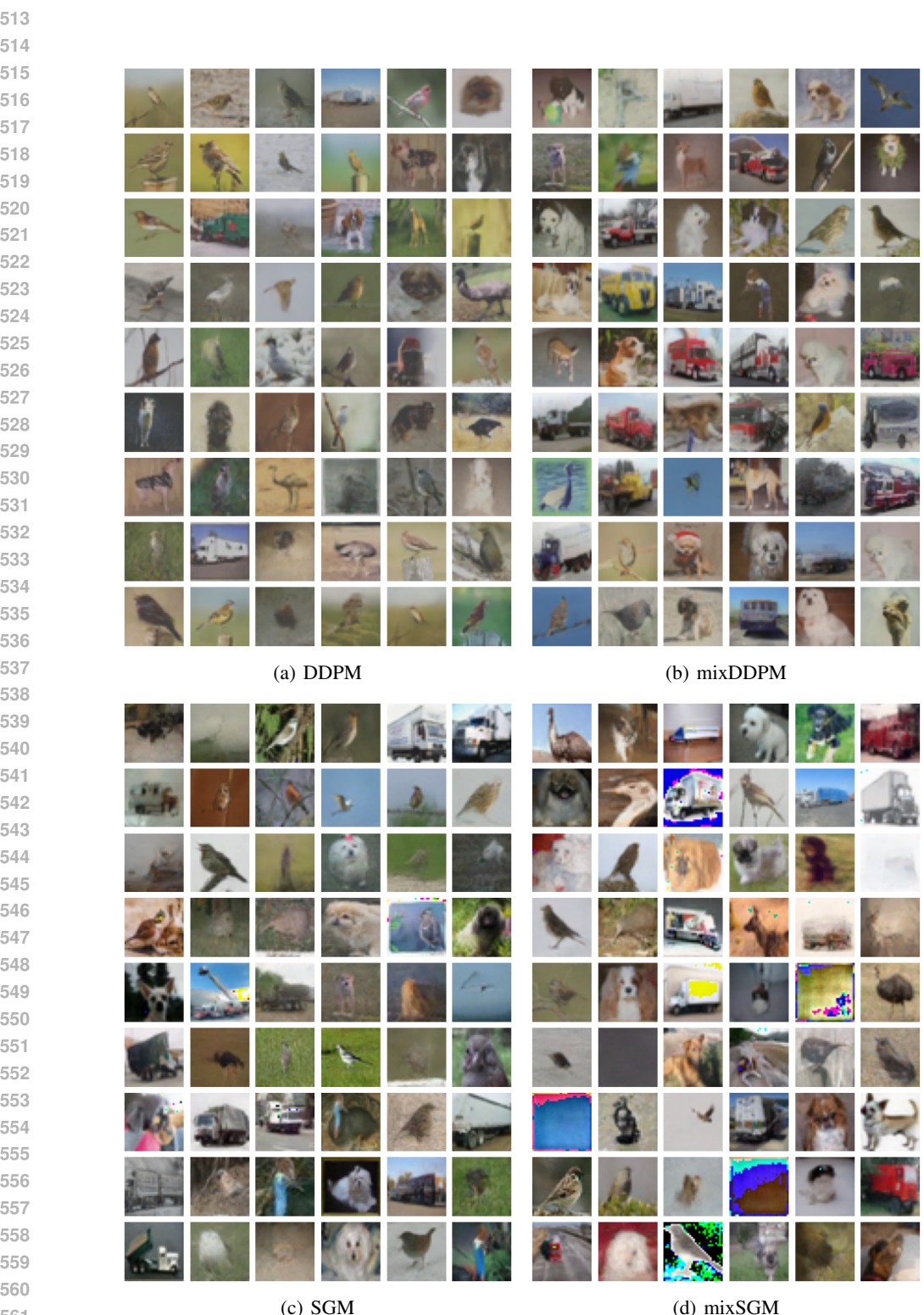

(a) DDPM             (b) mixDDPM

(c) SGM              (d) mixSGM

Figure 17: Experiments on CIFAR10 with 420k training steps

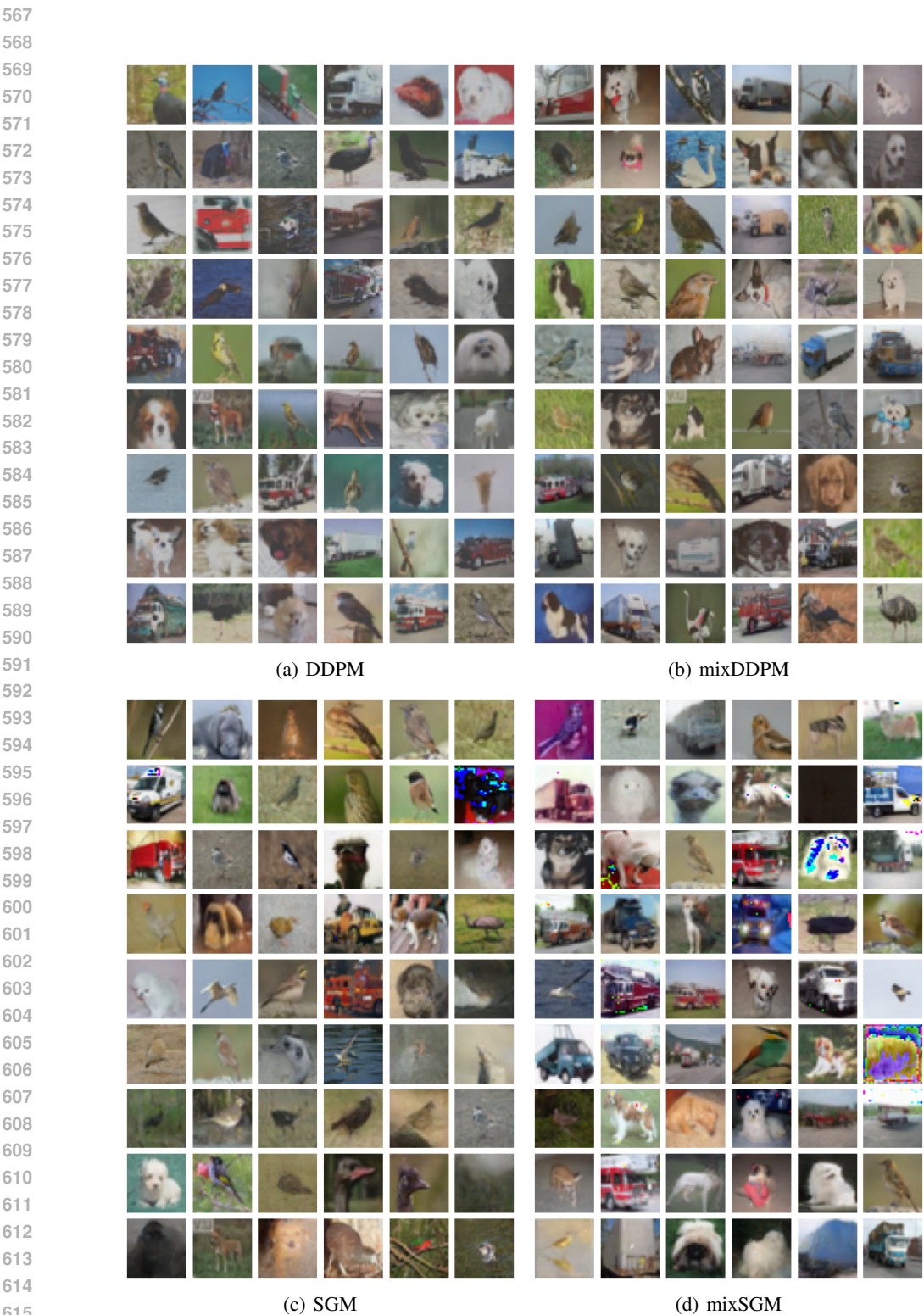

(a) DDPM                    (b) mixDDPM

(c) SGM                    (d) mixSGM

Figure 18: Experiments on CIFAR10 with 540k training steps

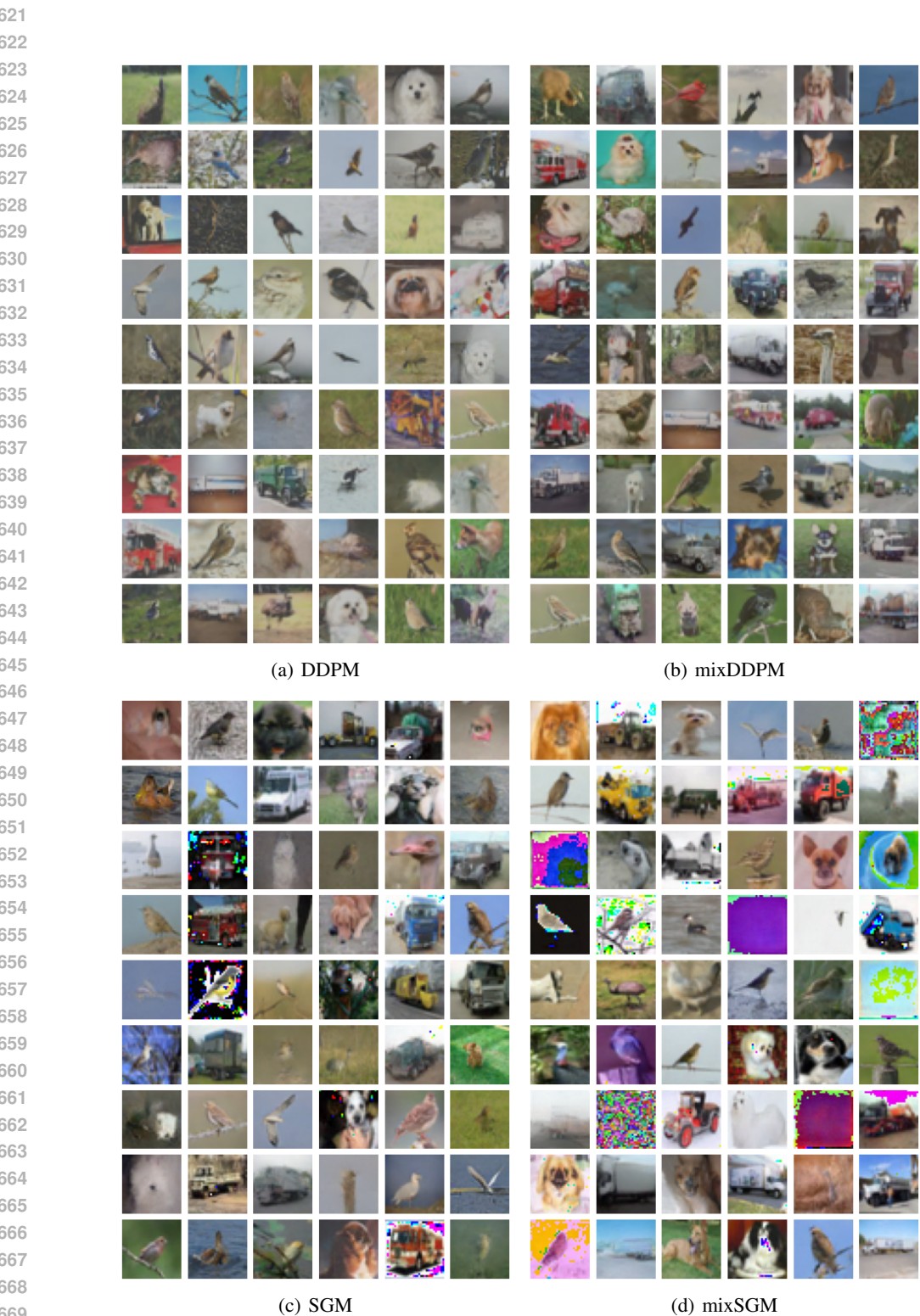

(a) DDPM             (b) mixDDPM

(c) SGM             (d) mixSGM

Figure 19: Experiments on CIFAR10 with 600k training steps

