# OpenReview forum: "Structured Diffusion Models with Mixture of Gaussians as Prior Distribution"
_ICLR.cc/2025/Conference — Submitted to ICLR 2025_

### Official Review · Reviewer_HhyJ · 2024-10-28

**Soundness:** 3
**Presentation:** 2
**Contribution:** 2
**Rating:** 3
**Confidence:** 3

**Summary:**

The paper presents a variant on diffusion models where the initial  noise (x_T)  has a Gaussian Mixture Model (GMM) distribution rather than a Gaussian distribution. The algorithm is equivalent to first clustering the datapoints, and then training a diffusion model separately for each cluster. The paper proves that by first clustering the points, the amount of work that the diffusion model has to do is lower. Experiments show that for a small number of iterations, the results of the proposed method have better FID than using a Gaussian prior.

**Strengths:**

The basic idea is simple and appealing: if the data distribution is composed of different clusters, it makes a lot of sense to model each cluster separately. This intuition is nicely formalized in proposition 1.

**Weaknesses:**

The algorithm is described quite poorly. At first reading, it looks very complicated but in fact the proposed method is just to first cluster the data and then run diffusion models separately on each cluster.

The requirement of first clustering the data is a weakness because finding a good clustering is in itself a difficult problem. I can imagine settings where getting a poor clustering will end up hurting the final performance.

In terms of significance, the paper mainly argues for lower effort (the average distance between the initial noise vector and the final generated sample). But it is not clear why in practice we would care about effort. The paper argues that this also leads to better visual results when we consider a limited number of training steps but I did not see any theoretical support for the improvement in visual quality. The numerical experiments also show better FID scores with a fixed number of training steps but the results seem inferior to previously published DDPM results on the same datasets.

**Questions:**

Is there a theoretical connection between effort and sample quality?

How do the samples that you generate with DDPM compare to published results?

---

> ### Author Response · Authors · 2024-11-26
>
> Below please find our responses to your questions.
>
> W1. The algorithm is described quite poorly. At first reading, it looks very complicated but in fact the proposed method is just to first cluster the data and then run diffusion models separately on each cluster.
>
> AW1. We appreciate the reviewer's recognition about the simplicity of our algorithm, which turns out to outperform classical diffusion models under the same training data, same overall parameter size and training resources.
>
> W2. The requirement of first clustering the data is a weakness because finding a good clustering is in itself a difficult problem. I can imagine settings where getting a poor clustering will end up hurting the final performance.
>
> AW2. If the user does not have additional information of the data distribution, selecting a large $K$ and then clustering can be time-consuming. In such cases, we recommend choosing a moderate $K$ rather than a large $K$. We have shown in the experiments that selecting a moderate $K$, such as $K = 2, 3$, is sufficient to outperform the $K = 1$ case. On the contrary, if the user has additional information about the data distribution, such as pre-given labels or clustering results in a low-dimensional subspace, they can determine $K$ based on their understanding of the data with fewer computation costs. Given $K$, the clustering step can be simplified with the user's understanding of the data distribution.
>
> W3. In terms of significance, the paper mainly argues for lower effort (the average distance between the initial noise vector and the final generated sample). But it is not clear why in practice we would care about effort. The paper argues that this also leads to better visual results when we consider a limited number of training steps but I did not see any theoretical support for the improvement in visual quality. The numerical experiments also show better FID scores with a fixed number of training steps but the results seem inferior to previously published DDPM results on the same datasets.
>
> AW3. From the theoretical view, we propose the reverse effort to demonstrate why mixed diffusion models with Gaussian Mixture prior can outperform classic diffusion models. When training resource is limited, the larger distance between prior distribution and data distribution may pose a challenge for the model to effectively move prior distribution to data distribution.
>
> From the empirical view, we admit that the samples we generated have larger FID scores compared to the state-of-the-art (SOTA) results of DDPM and SGM. We would like to clarify that our work does not aim to produce new SOTA results in diffusion models. Instead, our focus is on improving diffusion models with a Gaussian Mixture prior. We demonstrate through experiments that the mixed diffusion models outperform the classic diffusion models under the same experimental settings with limited training resources. The larger-than-SOTA FID score is a result of our effort to simulate a condition with limited training resources.
>
> Q1. Is there a theoretical connection between effort and sample quality?
>
> AQ1. To the best of our knowledge, there is no theoretical evidence that smaller effort can guarantee better sampling quality or smaller FID score. By introducing reverse effort, we aim to explain why the mixed diffusion models outperform the classic diffusion models by connecting the prior distributions and the data distribution.
>
> Q2. How do the samples that you generate with DDPM compare to published results?
>
> AQ2. The samples we generated exhibit larger FID scores compared to the state-of-the-art (SOTA) results of DDPM and SGM. We would like to clarify that our work does not aim to produce new SOTA results in diffusion models. Instead, our focus is on improving diffusion models with a Gaussian Mixture prior. We demonstrate that the mixed diffusion models outperform the classic diffusion models under the same experimental settings with limited training resources. The larger-than-SOTA FID score is a result of our effort to simulate a condition with limited training resources.
>
> In our latest version, we have corrected several typos and added citations for some related literature. Additionally, we have included Appendix A to introduce variance estimation for mixSGM, Appendix B to visualize the reverse process and demonstrate how a smaller reverse effort improves sample quality, and Appendix E.1 to present the experimental results on CelebA-HQ, a high-resolution image dataset.

---

### Official Review · Reviewer_jnSw · 2024-10-31

**Soundness:** 2
**Presentation:** 2
**Contribution:** 1
**Rating:** 5
**Confidence:** 4

**Summary:**

Diffusion models have become prominent generative models for tasks like image and audio synthesis, but they demand high computational resources due to their multistep noise-corruption and denoising processes. This intensive training can pose challenges, especially for users with limited resources, like small enterprises or real-time applications, where accelerated training is essential. To address these constraints, the authors propose enhancing classical diffusion models by adjusting the prior distribution. Instead of the standard Gaussian prior, they suggest using a Gaussian Mixture Model (GMM) as the prior, which can leverage any inherent structure in the data. This approach aims to improve training efficiency and maintain model performance even with fewer resources. Key contributions of this work include:

(1) Developing “mixed diffusion models” that use a GMM prior, detailing both forward and reverse processes, and introducing variants like Mixed Denoising Diffusion Probabilistic Models (mixDDPM) and Mixed Score-based Generative Models (mixSGM).

(2) Introducing a new metric, “Reverse Effort,” to measure efficiency by quantifying the reduction in effort needed during the reverse denoising process.

(3) Demonstrating the model’s efficiency through experiments on synthetic and real datasets, highlighting its effectiveness under limited training resources.

**Strengths:**

1. Perhaps the introduction of a “Reverse Effort” metric provides a fresh perspective on quantifying the model's efficiency and aligns closely with practical concerns around computational cost in generative modeling.

2. The paper is written with clarity and conciseness, making complex concepts accessible. The structure follows a logical progression, starting with motivation, theoretical developments, and concluding with empirical validation.

**Weaknesses:**

1. The choice of a Gaussian Mixture Model (GMM) as the prior is intriguing, but the paper does not fully analyze why a GMM is particularly well-suited for diffusion models over other structured priors (e.g., Student’s t-distribution, other hierarchical models).

2. While the “Reverse Effort” metric is an interesting addition, its practical interpretation and relevance could be better explained. The paper might improve by providing additional intuition behind this metric, perhaps through visualizations or step-by-step examples demonstrating how “Reverse Effort” correlates with sample quality.

3. This reviewer has certain concerns over the model structure based on the mixture of Gaussian prior. As both diffusion forward and reserve processes are designed based on the cluster label information of the data which is pre-determined according to the initial centroids, thus the entire process is roughly equivalent to couple of individual diffusion process (although all the noise models share the same networks and trained individually).  Originally I would expect that the mixture of Gaussian structure can be propagated on the chain. But it is not.  This entirely downgrades the paper's innovation level.

**Questions:**

1. It seems the process depends on the initialization of the centroids c_j.   Could the author(s) explore more on this initialization.  What is its impact on the final generalized distribution.

2. Several experiments have been conducted on several basic datasets such as EMNIST and Cifar10.  They are quite classic.  Can more modern dataset be used for testing, for example, LSUN (Large-scale Scene UNderstanding Challenge)?

3. Why in Algorithm 2, the cluster index j is gone missing in noise model \epsilon_{\theta}?

4. Perhaps ReEff defined in (14) and (15) are OT-distance?

---

> ### Author Response · Authors · 2024-11-26
>
> Thank you for your valuable feedback and helpful suggestions. Below please find our responses to your questions.
>
> W1. The choice of a Gaussian Mixture Model (GMM) as the prior is intriguing, but the paper does not fully analyze why a GMM is particularly well-suited for diffusion models over other structured priors (e.g., Student’s t-distribution, other hierarchical models).
>
> AW1. We agree with the reviewer that diffusion models with various structured priors, including Student's t-distribution and other hierarchical models, have been studied. The choice of the Gaussian Mixture Model (GMM) is not inherently superior to other possible choices. For instance, Pandey et al. (2024, arXiv) introduce diffusion models with Student's t-distribution as a prior to estimate heavy-tail distributions in weather datasets. Skorokhodov et al. (2024, arXiv) implement hierarchical patch diffusion models for high-resolution video generation.
>
> The GMM prior in our work offers the following benefits. First, we add Gaussian noise during the forward process, retaining a similar form to that of classic diffusion models. This enables us to make minimal code changes during training. Second, the reverse process in our model has either a Gaussian transition kernel (mixDDPM) or an SDE representation (mixSGM). This allows the model to invoke the neural network only once per step and avoid using Langevin dynamics, which require multiple neural network calls at each step. Finally, GMM provides additional convenience for theoretical analysis. The reverse effort, as defined in our work, relies on the existence of the second-order moments of the prior. If the second-order moment of the prior does not exist or is infinite, the reverse effort is not well-defined and cannot illustrate why mixed diffusion models outperform classic diffusion models.
>
> W2. While the “Reverse Effort” metric is an interesting addition, its practical interpretation and relevance could be better explained. The paper might improve by providing additional intuition behind this metric, perhaps through visualizations or step-by-step examples demonstrating how “Reverse Effort” correlates with sample quality.
>
> AW2. Inspired by the reviewer's comments, we have added Appendix B in the latest version. In Appendix B, we demonstrate how a smaller reverse effort helps the model reveal clusters more quickly and leads to better sample quality through visualizations of the reverse process in 2D experiments. Please correct us if we misunderstood your comments.
>
> W3. This reviewer has certain concerns over the model structure based on the mixture of Gaussian prior. As both diffusion forward and reserve processes are designed based on the cluster label information of the data which is pre-determined according to the initial centroids, thus the entire process is roughly equivalent to couple of individual diffusion process (although all the noise models share the same networks and trained individually). Originally I would expect that the mixture of Gaussian structure can be propagated on the chain. But it is not. This entirely downgrades the paper's innovation level.
>
> AW3. We acknowledge that not incorporating an adaptive prior is a limitation of this work. Initially, we attempted to design a method to integrate the centers into the neural network so that they could be updated during training. However, this approach faced a challenge, as the neural network is trained to predict noise rather than the centers. We leave the development of adaptive priors as future work and anticipate its potential success.

---

> ### Author Response · Authors · 2024-11-26
>
> Q1. It seems the process depends on the initialization of the centroids $c_j$. Could the author(s) explore more on this initialization. What is its impact on the final generalized distribution.
>
> AQ1. We agree with the reviewer that the model efficiency is dependent on the initialization of the centers $c_j$. A poor initialization typically leads to degraded sample quality. We also recommend selecting the centers as the arithmetic mean of all the data points allocated to that center. Our propositions regarding the reduction of reverse efforts are based on this assumption.
>
> Q2. Several experiments have been conducted on several basic datasets such as EMNIST and Cifar10. They are quite classic. Can more modern dataset be used for testing, for example, LSUN (Large-scale Scene UNderstanding Challenge)?
>
> AQ2. We appreciate your kind advice on conducting experiments on more modern datasets. Inspired by your comments, we have added experiments on CelebA-HQ, a high-resolution dataset consisting of facial images, in the latest version. The numerical results indicate that the mixed diffusion models outperform the classic diffusion models in high-resolution image generation. Please refer to Section E.1 for more details.
>
> Q3. Why in Algorithm 2, the cluster index j is gone missing in noise model $\epsilon_{\theta}$?
>
> AQ3. Thank you for pointing out our typo. We have corrected it in the latest version.
>
> Q4. Perhaps ReEff defined in (14) and (15) are OT-distance?
>
> AQ4. To the best of our knowledge, the reverse effort does not correspond to the Optimal Transport (OT) distance. The OT distance is defined as the infimum of the distance between two distributions over all possible couplings. In contrast, our reverse effort is defined as the $l_2$ distance between the data distribution and the prior distribution, where the coupling is specifically induced by the forward process. This predefined coupling does not guarantee that the resulting distance achieves the infimum over all possible couplings.
>
> In our latest version, we have corrected several typos and added citations for some related literature. Additionally, we have included Appendix A to introduce variance estimation for mixSGM, Appendix B to visualize the reverse process and demonstrate how a smaller reverse effort improves sample quality, and Appendix E.1 to present the experimental results on CelebA-HQ, a high-resolution image dataset. Thank you again for your helpful comments.

---

> ### Comment · Reviewer_jnSw · 2024-11-26
>
> Thank you for your clarification and rebuttals.
>
> Having said that, I still feel the basic idea of using individual Gaussian diffusion models does not bring much benefit from complex data distributions, particularly the initialization and number of cluster issues.  Given this, I may not raise my current score but I dont objection a possible acceptance.

---

> > ### Author Response · Authors · 2024-11-29
> >
> > Thank you for taking the time and efforts to review our work and responses. We appreciate your feedback and the openness of a possible acceptance.
> >
> > We agree that that the clustering problem itself inserts challenges to the use of our model, when the original data lack a clear structure of clusters and the user does not have domain knowledge of the data clusters. Our work may better fit applications where the data cluster structure is relatively clear in some dimensions, or user has some prior domain knowledge about the data structure.
> >
> > Thanks again for your valuable and helpful insights.

---

### Official Review · Reviewer_XkiZ · 2024-11-03

**Soundness:** 4
**Presentation:** 4
**Contribution:** 2
**Rating:** 5
**Confidence:** 3

**Summary:**

The paper studies a class of structured diffusion models, mixDDPM and mixSGM, using a Gaussian Mixture Model (GMM) as the base distribution (prior distribution) instead of Gaussian, in order to utilize more structure information of the dataset.

**Strengths:**

Instead of using Gaussian prior as most work does, the authors make use of the potential clustering information from the training set with GMM priors. This improves the efficiency for the cluster-finished problems. The authors also discuss practical implementations, such as using a data-driven clustering method to select the centers of the GMM. The experiments on datasets like MNIST show some improvements training steps. The authors also give simple computational algorithms.

**Weaknesses:**

0. Innovation. Please notice the paper: Guo et al (2023, NeurIP 2023) and Zach et al (2023, SSVM). The ideas seem quite similar. Moreover, there are a lot of work using non-Gaussian priors, or actually more general priors.
1. Although the authors mention 'not all scenarios where diffusion models are used enjoy access to extensive training re-sources', the scale of the experiments are far too small. Have you tried scaling up?
2. What if the number of clusters are quite large? Solving large clusters problem can be another new issues. If having numerous number of clusters, the prior can be ill-posed. It can be close to nonparametric case and the computation is quite expensive. This method already restricts the dim of the problem. It seems that the number of clusters also needs restricted.
3. Theoretically, from Gaussian to GMM, people just need one step, using one more layer like in the hierarchical models.
4. CIFAR-10 and NIST datasets are a little bit small. And in the experiment, only original DDPM and SGM compared. FIDs are far higher than SOTA, maybe indicating the lack of training.

**Questions:**

Please see the weaknesses.

---

> ### Author Response · Authors · 2024-11-26
>
> Thank you for your valuable feedback and helpful suggestions. Below please find our responses to your questions.
>
> Q0. Innovation. Please notice the paper: Guo et al (2023, NeurIP 2023) and Zach et al (2023, SSVM). The ideas seem quite similar. Moreover, there are a lot of work using non-Gaussian priors, or actually more general priors.
>
> A0. We thank the reviewer for pointing out these relevant papers. Guo et al. (2023, NeurIPS 2023) focus on utilizing the Gaussian Mixture Model (GMM) as transition kernels for the reverse process. In their model, the priors of the reverse process remain Gaussian. Zach et al. (2023, SSVM) model the marginal density of the forward process as a product of one-dimensional GMMs under certain conditions. Their priors are neither Gaussian nor Mixed Gaussian. In contrast to these works, our approach retains the Gaussian noise while modifying both the forward and reverse processes to achieve a GMM prior to improve model efficiency. We also agree with the reviewer that other works explore non-Gaussian priors. Benefiting from the Gaussian noise, the reverse processes of our models take a form similar to that of the original diffusion models (DDPM and SGM). This allows our model to avoid repeatedly using Langevin dynamics during the reverse process, thereby maintaining relatively faster sampling speeds.
>
> Q1. Although the authors mention 'not all scenarios where diffusion models are used enjoy access to extensive training re-sources', the scale of the experiments are far too small. Have you tried scaling up?
>
> A1. We appreciate the reviewer for pointing out this issue. We also acknowledge that the scope of our experiments was relatively small. Our initial design was to test the mixed diffusion models on small-scale experiments with the expectation that they would perform well on larger scales. Inspired by your question, we have scaled up our experiments on CelebA-HQ, a larger dataset consisting of facial images. The numerical results consistently demonstrate that the mixed diffusion models outperform the classic diffusion models. For more details, please refer to Appendix E.1 of the latest version.
>
> Q2. What if the number of clusters are quite large? Solving large clusters problem can be another new issues. If having numerous number of clusters, the prior can be ill-posed. It can be close to nonparametric case and the computation is quite expensive. This method already restricts the dim of the problem. It seems that the number of clusters also needs restricted.
>
> A2. We agree with the reviewer that solving large clusters problem to select the centers can become time-consuming if $K$ is large and the user does not have additional information about the data distribution. In such cases, we recommend choosing a moderate $K$ rather than an excessively large $K$. We have shown in the experiments that selecting a moderate $K$, such as $K = 2, 3$, is sufficient to outperform the $K = 1$ case. On the contrary, if the user has additional information about the data distribution, such as pre-given labels or clustering results in a low-dimensional subspace, they can determine $K$ based on their understanding of the data with fewer computation costs. Once $K$ and the centers are chosen, the computational effort required for training and sampling the mixed diffusion models is orthogonal to the center selection process. Please correct us if we misunderstood the question.
>
> Q3. Theoretically, from Gaussian to GMM, people just need one step, using one more layer like in the hierarchical models.
>
> A3. By "hierarchical model", we suppose the reviewer is referring to "Bayesian hierarchical model". Yes, the mixed diffusion models can be understood as a class of special Bayesian hierarchical models, just like diffusion models can be understood as a sampling method. We want to note that the intuition behind the "one step" comes from the observation that general Gaussian priors may fail to capture the data structure and GMM priors can help reduce the effort during the reverse process.

---

> ### Author Response · Authors · 2024-11-26
>
> Q4. CIFAR-10 and NIST datasets are a little bit small. And in the experiment, only original DDPM and SGM compared. FIDs are far higher than SOTA, maybe indicating the lack of training.
>
> A4. We appreciate the reviewer for pointing out the lack of numerical experiments on larger datasets. Inspired by your comments, we have now added experiments on CelebA-HQ, a high-resolution dataset consisting of facial images. The results indicate that the mixed diffusion models outperform the classic diffusion models in high-resolution image generation. For more details, please refer to Appendix E.1 in the latest version.
>
> Due to page limitations, we have only compared the classic DDPM and SGM. Other types of diffusion models can also be extended to the mixed diffusion models. For instance, the classic DDIM follows the forward process:
> \begin{equation*}
>     x_t = \sqrt{\alpha_t} x_0 + \sqrt{1 - \alpha_t} \epsilon,
> \end{equation*}
> where $\epsilon\sim\mathcal{N}(0,I)$. Suppose that $x_0$ is assigned to the $j$-th center $c_j$. The forward process for the mixDDIM can be written as
> \begin{equation*}
>     x_t = \sqrt{\alpha_t} (x_0-c_j) + \sqrt{1 - \alpha_t} \epsilon+c_j.
> \end{equation*}
> In addition, the initial reverse process of the DDIM is given by
> \begin{equation*}
>    x_{t-1} = \sqrt{\alpha_{t-1}}\frac{x_t - \sqrt{1 - \alpha_t} \epsilon_\theta(x_t, t)}{\sqrt{\alpha_t}}  + \sqrt{1 - \alpha_{t-1}} \epsilon_t.
> \end{equation*}
> For the mixDDIM, the reverse process can be formalized as
> \begin{equation*}
>    x_{t-1} = \sqrt{\alpha_{t-1}}\frac{x_t -c_j- \sqrt{1 - \alpha_t} \epsilon_\theta(x_t, t,j)}{\sqrt{\alpha_t}}  + \sqrt{1 - \alpha_{t-1}} \epsilon_t+c_j.
> \end{equation*}
>
> We want to clarify that we are not proposing new state-of-the-art (SOTA) diffusion models. Rather, our aim is to demonstrate that the mixed diffusion models outperform the classic diffusion models under limited training resources. The higher-than-SOTA FID scores observed in our results are a consequence of our efforts to simulate scenarios with constrained training resources.
>
> In our latest version, we have corrected several typos and added citations for some related literature. Additionally, we have included Appendix A to introduce variance estimation for mixSGM, Appendix B to visualize the reverse process and demonstrate how a smaller reverse effort improves sample quality, and Appendix E.1 to present the experimental results on CelebA-HQ, a high-resolution image dataset. Thank you again for your helpful comments.

---

### Official Review · Reviewer_VbvB · 2024-11-04

**Soundness:** 2
**Presentation:** 3
**Contribution:** 2
**Rating:** 5
**Confidence:** 3

**Summary:**

The paper presents a class of structured diffusion models that utilize a mixture of Gaussian distributions as the prior, rather than the conventional standard Gaussian distribution. It also proposes a straightforward training procedure that effectively integrates this mixed Gaussian prior. Through numerical experiments with the EMNIST and CIFAR10 datasets, the authors demonstrate that the proposed mixDDPM and mixSGM achieve better generation performance than DDPM and SGM at the same sampling steps.

**Strengths:**

1. The structure of the paper is well organized and effectively facilitates the presentation of its content.
2. It is both intuitive and reasonable to consider employing a mixed Gaussian prior in diffusion models, which, as opposed to a single Gaussian distribution, allows for a more accurate representation of the real data distribution. The resulting performance improvements align well with this intuition.

**Weaknesses:**

1. The paper lacks a thorough discussion of the impact of varying K values on model performance, particularly whether mixDDPM merely degrades to DDPM when K=1 or if it has a more efficient training procedure compared to DDPM. Additionally, it fails to address whether an excessively large K introduces unnecessary computational overhead or whether the choice of K selection algorithms affects the results.
2. While the experimental results demonstrate that mixDDPM achieves better generation performance than DDPM with the same number of steps, the paper does not address whether the combination of multiple Gaussian distributions incurs greater storage and computational overhead during model training and whether there is a trade-off involved.
3. The paper may need to include additional experiments. It would be important to investigate whether mixDDPM and mixSGM remain applicable to more complex data structures, such as high-resolution image generation scenarios.
4. The manuscript contains several typos, such as in Section 4.2, where the first sentence reads, "Suppose a given data x0is assigned," missing a space between "x0" and "is."

**Questions:**

See the weaknesses part above.

---

> ### Author Response · Authors · 2024-11-26
>
> Thank you for your valuable feedback and helpful suggestions. Below please find our responses to your questions.
>
> Q1. The paper lacks a thorough discussion of the impact of varying K values on model performance, particularly whether mixDDPM merely degrades to DDPM when K=1 or if it has a more efficient training procedure compared to DDPM. Additionally, it fails to address whether an excessively large K introduces unnecessary computational overhead or whether the choice of K selection algorithms affects the results.
>
> A1. When $K=1$ and $c_1=0$, the mixDDPM reduces to the classic DDPM. For excessively large $K$, the computational effort may increase in two main aspects: the selection of the centers $c_1, \cdots, c_K$, and the training and sampling of the mixDDPM after the centers have been selected. The selection of centers can become time-consuming if $K$ is large and the user does not have additional information about the data distribution. In such cases, we recommend choosing a moderate $K$ rather than an excessively large $K$. In our experiments, we demonstrated that selecting a moderate $K$, such as $K = 2, 3$, is sufficient to outperform the $K = 1$ case. On the contrary, if the user has additional information about the data distribution, such as pre-given labels or clustering results in a low-dimensional subspace, they can determine $K$ based on their understanding of the data. Once $K$ and the centers are chosen, the computational effort required for training and sampling the mixDDPM is orthogonal to the center selection process. Regarding the UNet model, increasing the number of classes introduces only a minor increase in model size and computational cost compared to the backbone UNet itself.
>
>
> We also agree with the reviewer that the choice of $K$-selection algorithms affects the results.
>
>
> Q2. While the experimental results demonstrate that mixDDPM achieves better generation performance than DDPM with the same number of steps, the paper does not address whether the combination of multiple Gaussian distributions incurs greater storage and computational overhead during model training and whether there is a trade-off involved.
>
> A2. Regarding the neural network, the mixed diffusion models require only minor additional storage since the $K$ classes in the UNet add little storage compared to the UNet itself. Compared to the training of DDPM, the training of mixDDPM involves specifying a center for a given data point before adding noise to it. This process can be slow if $K$ is excessively large. To accelerate this process, we recommend manually adding labels to each data point specifying the centers for the data before the entire training process. From the experiments, we observe that mixDDPM requires approximately $0.04\\%$ more storage and $5\\%$ more training time for the same number of epochs.
>
>
> Q3. The paper may need to include additional experiments. It would be important to investigate whether mixDDPM and mixSGM remain applicable to more complex data structures, such as high-resolution image generation scenarios.
>
> A3. Inspired by the reviewer's comment, we have added two experiments, which we will introduce below.
>
> First, we extend the mixSGM to incorporate variance estimation. Instead of using $\mathcal{N}(c_j,I)$ as the $j$-th component of the prior, we extend it to $\mathcal{N}(c_j,\sigma_j^2I)$. The additional parameter $\sigma_j$ can be estimated from data distribution. By multiplying the $\sigma_j$ to the diffusion scaler in both the forward and the reverse SDE, we improve the original mixSGM. More details of the method and the numerical results can be found in Appendix A in the latest version. In general, we have achieved significant improvement in terms of FID score compared to the original mixSGM.
>
> Second, we implement the mixed diffusion models on CelebA-HQ, a high-resolution dataset on facial images. Numerical results consistently suggest that the mixed diffusion models have outperformed the classic diffusion models. Please refer to Appendix E.1 in the latest version for generated images.
>
>
> Q4. The manuscript contains several typos, such as in Section 4.2, where the first sentence reads, "Suppose a given data x0is assigned," missing a space between "x0" and "is."
>
> A4. We thank the reviewer for pointing out our typos. We have corrected them in the latest version.
>
>
> In our latest version, we have corrected several typos and added citations for some related literature. Additionally, we have included Appendix A to introduce variance estimation for mixSGM, Appendix B to visualize the reverse process and demonstrate how a smaller reverse effort improves sample quality, and Appendix E.1 to present the experimental results on CelebA-HQ, a high-resolution image dataset. Thank you again for your helpful comments.

---

### Meta-Review · Area_Chair_Uime · 2024-12-20

**Metareview:**

The paper proposes a class of structured diffusion models, in which the prior distribution is chosen as a mixture of Gaussians, rather than a standard Gaussian distribution. The proposed formulation is accompanied by numerical experiments.

Reviewers generally agree that it is reasonable and natural to consider Gaussian mixtures as prior distribution as they can better capture real data distributions. However, there are concerns about the innovation level of the proposed approach and its practical performance.

**Additional Comments On Reviewer Discussion:**

- Reviewer HhyJ:  The proposed method is just to first cluster the data and then run diffusion models separately on each cluster. However, good clustering is in itself a difficult problem and a poor clustering can hurt the final performance. The authors responded by recommending a moderate cluster number K rather than a large K (e.g. K=2,3 in their experiments), however this is only a heuristic.

- Reviewer  jnSw raised similar concerns: The basic idea of using individual Gaussian diffusion models does not bring much benefit from complex data distributions, particularly with the initialization and number of cluster issues.

- Reviewer XkiZ: the reported results are inferior to state of the art (SOTA). The authors responded that their aim is not to have a SOTA model, but to show that their method outperforms the classic diffusion models under limited training resources. This raises the issue of whether the proposed model brings much additional benefit with sufficient training (echoing the previous two concerns).

These points, among others, lead to the reject decision of the paper.

---

### Decision · Program_Chairs · 2025-01-22

Reject